# Secondary acceleration of slip fronts driven by slow slip event coalescence in subduction zones

Ji Wang[1,2], Kejie Chen [1,3] ✉, Sylvain Michel [4,5,6], Luca Dal Zilio [7,8], Hai Zhu[1], Lei Xia[1], Jun Xie[1] & Shunqiang Hu[9]

The coalescence of slow slip events (SSEs) in subduction zones has been proposed as a potential precursor to large earthquakes, yet the physical conditions under which SSE fronts coalesce remain poorly understood. Here, we investigate coalescing SSEs along the Cascadia subduction zone. Using Global Navigation Satellite System (GNSS) data, we invert for the spatiotemporal evolution of the slip rate of SSEs from 2012 to 2023. We identify a coalescing event in 2021, which occurred during a phase of SSEs moment rate decrease, contrary to the previously documented 2013 coalescence. Coalescence triggered a secondary increase in slip rate and a rupture expansion in the 2021 event. To explore the mechanisms driving coalescence, we perform numerical simulations based on rate-and-state friction. Our results show that heterogeneity in frictional parameters and effective normal stress influences the occurrence rate and slip rate evolution of coalescing SSEs by modulating their propagation speeds and interaction probabilities. Although coalescing events lack distinct moment–duration or moment–area scaling trends, they are part of the broader class of migrating SSEs, which are associated with a b-value change in the magnitude–frequency distribution. These findings improve our understanding of SSE coalescence, which can potentially influence the timing and extent of future earthquakes.

Slow slip events (SSEs) represent an intermediate class of fault slip, occurring between the rapid rupture speeds of earthquakes and the steady interseismic loading of tectonic plates, and thus play a pivotal role in the seismic cycle[1–3]. SSEs typically progress through several stages: nucleation, along-dip saturation, along-strike acceleration and deceleration of the propagation of the rupture front (i.e., the boundaries of the slipping region) and slip rate[4]. These stages together contribute to the complex dynamics of SSEs, which can vary depending on local fault conditions and, in some cases, the interaction or coalescence of multiple SSE fronts (see Fig. S1). The coalescence of rupture fronts, also called the collision, merging, or focusing effect, occurs when two ruptures propagate toward each other and merge[5]. Coalescence at different SSE stages yields distinct slip characteristics (Fig. S1). In subduction zones such as Cascadia, SSEs are recurrently

¹Department of Earth and Space Sciences, Southern University of Science and Technology, Shenzhen, China. ²Shanghai Innovation Institute, Shanghai, China. ³Key Laboratory of Earthquake Forecasting and Risk Assessment, Ministry of Emergency Management, Southern University of Science and Technology, Shenzhen, China. ⁴Université Côte d'Azur, IRD, CNRS, Observatoire de la Côte d'Azur, Géoazur, Valbonne, France. ⁵CNRS-INSU, Institut des Sciences de la Terre Paris, Sorbonne Université, ISTeP UMR 7193, Paris, France. ⁶Laboratoire de Géologie, Département de Géosciences, Ecole Normale Supérieure, PSL Université, Paris, France. ⁷Earth Observatory of Singapore, Nanyang Technological University, Singapore, Singapore. ⁸Asian School of the Environment, Nanyang Technological University, Singapore, Singapore. ⁹Key Laboratory of Poyang Lake Wetland and Watershed Research, Ministry of Education, Jiangxi Normal University, Nanchang, China. ✉e-mail: chenkj@sustech.edu.cn

observed through geodetic measurements and are frequently accompanied by tectonic tremor, jointly forming episodic tremor and slip (ETS)[6,7]. ETS events transiently release accumulated stress. This process may increase stress at the down-dip limit (25–45 km) of the seismogenic locked zone, potentially modulating the rupture area of future megathrust earthquakes.

Despite extensive observational data, the underlying mechanics of SSEs–particularly the conditions that drive slow slip towards faster rupture speeds–remain incompletely understood[8]. Previous observations[9] from northern Cascadia reveal that the coalescence of two initially separated SSEs can substantially increase both slip rate and final moment release, implying that SSE coalescence may accelerate fault slip. This is corroborated by laboratory friction experiments, which show that abrupt changes in slip rate can modify fault strength, thereby activating fault-weakening mechanisms that may trigger either slow or rapid fault motion[2]. However, the frequency and impacts of SSE coalescence across different subduction environments remain uncertain. The dynamic behaviour of SSEs is further complicated by fault stress memory along the subduction interface, which may govern SSE initiation, propagation, and eventual coalescence[8,10]. Fault heterogeneity (e.g., variations in effective normal stress and frictional properties) not only affects stress distribution along the fault but also controls slip fronts interactions during coalescence, potentially influencing SSE duration and moment release[11].

Here, we focus on a complex SSE front coalescing event observed in 2021 beneath Southern Oregon and Northern California. This event provides a rare opportunity to examine the interaction between coalescing SSE fronts and their implications in terms of slip acceleration. Unlike the 2013 SSE coalescing event[9], which occurred during the acceleration phase when the rupture fronts of both SSEs were propagating with increasing speed, the 2021 event occurred during the deceleration phase, when rupture fronts of both SSEs were propagating with decreasing speed (Fig. S1). Using a deep learning-based SSEs detection method and kinematic inversion, we quantify slip rate variations and explore the relationship between slip, stress evolution, and tremor activity throughout the coalescing process. Furthermore, we introduce a simplified width-bounded rate-and-state model with along-strike heterogeneity to examine the characteristics and controlling factors of SSE coalescence. Finally, we discuss how SSE rupture migration, coalescence, and expansion influence their moment release and frequency–magnitude distribution characteristics.

## Results

### The secondary peak in the moment rate from slip coalescence

In this study, we examine the evolution of slip rates associated with SSEs from 2012 to 2022 through kinematic inversions of GNSS data. To identify the timing of each SSE and the corresponding affected GNSS stations, we use detection results from a previous study[12], which employed a machine learning model (see "Methods" for details). This catalogue serves as a reference for kinematic inversions, enabling to investigate slip rate characteristics for each event. Here, we focus on a complex SSE that occurred between March and May 2021, spanning latitudes 40°N to 46°N. This event is noteworthy because it occurred during a deceleration phase, in contrast to the acceleration phase of the 2013 event.

The SSE released energy equivalent to a moment magnitude (*Mw*) 6.6 earthquake. Based on the spatial continuity in the evolution of the slip distribution, we segmented the event into two phases: slip activity preceding April 14 (designated SSE1) and the subsequent phase (designated SSE2) (Fig. 1a). SSE1 occurred predominantly in the southern portion of the region at depths of 20–70 km (Fig. S9). Starting on April 15, SSE1 exhibited a typical along-strike northward migration at an average propagation rate of ~7.4 km/day. The moment release rate peaked by April 8 before subsiding.

The coalescence of slip fronts occurred during SSE2 (Fig. 1b). After April 14, the slip area associated with SSE1 diminished, while a new slip

area (SSE2.a) emerged at latitude 44.8°N, accompanied by a burst of tremor activity. SSE2.a reached a peak in moment release rate on April 23 ($1.85 \times 10^{17}$ N·m/day), coinciding with the emergence of a new slip area (SSE2.b) at latitude 43.1°N, in continuity of SSE1. SSE2.a then began to migrate southward along-strike at a speed of ~13.4 km/day. On April 26, the moment release rate of SSE2.b peaked ($1.95 \times 10^{16}$ N·m/day), after which both SSE2.a and b entered a deceleration phase in moment rate. On April 29, the slip fronts coalescence triggered a new phase of renewed increase in moment release rate. This coalescing produced a secondary peak in moment release rate on May 4, eight days after the onset of the coalescence, before the slip rate resumed its decline.

Bletery and Nocquet[9] noted a similar pattern, in which moment release rates surged over 2–5 days after the onset of the coalescence phase in 2013 SSE. For the 2013 event, slip fronts coalesced during the slip rate acceleration phase of both SSEs, and this interaction further enhanced the slip acceleration, resulting in slip bursts[8,13]. For the 2021 event, each SSE exhibits distinct front propagation along strike before coalescence. During this phase, we can analyse both front migration speeds and the corresponding slip rate changes. However, once the two SSEs coalesce, the slipping zones unify, and the notion of two separate slip fronts no longer applies in a meaningful way. At this stage, we observe a secondary increase in slip rate and moment release, accompanied by renewed rupture area expansion. Although this expansion can be described as a new front propagation, it is no longer related to the original two SSE fronts. The coalescence of slow slip fronts provides a mechanism that counteracts slip damping[9]. This renewed acceleration phase manifests as a second peak in the SSE source time function, marking a distinct stage in SSE progression.

### Slip fronts coalescence promotes tremor occurrence

The strong correlation between geodetically estimated moment release and tremor activity suggests that tectonic tremors can serve as a reliable proxy for tracking SSEs[14,15]. For SSE2, distinct peaks in tremor count align with the two major peaks in moment release rate, with an additional minor peak observed during the coalescence of slip fronts (Fig. 1a). This observation suggests that the coalescence of slip fronts enhanced tremor activity. During the coalescing period, increased stress may trigger additional tremor occurrences due to slip acceleration[2].

We further investigate the SSEs coalescing process by comparing the along-strike relationship among tremor activity, slip rate, and stress release rate along the 36 km iso-depth (representing the central depth of the slip zone) within a 10 km-wide swath from April 27 to May 2 (Fig. 2a). We specifically calculate shear stresses aligned with the slip direction using the EDCMP program[16]. The analysis shows that, before the coalescing event, tremors were predominantly concentrated at the leading edge of the SSE2.b slip front, consistent with the hypothesis that tremors preferentially occur at sites of high slip rate during slip migration.

During the onset of slip coalescence, two slip fronts converge in the coalescing zone, leading to a peak in tremor activity. This peak is driven by elevated slip rates and the combined shear stress transients from both slip fronts, which are sufficient to trigger tremors due to their high sensitivity to minor stress changes (e.g., 1–2 kPa/day, comparable to tidal or teleseismic stresses)[17,18]. Tremors are predominantly located within the coalescing zone, positioned between the two regions of maximum stress release (Fig. 2b, c). As the coalescing process progresses, slip rate in the coalescing zone gradually increases, while stress release rates continue to rise. Tremor counts decreased by 36% relative to the peak between April 29 and 30 (Fig. 2a), but then roughly stabilized over the following days. The decline occurs because tremors reach a seismic saturation point after releasing ~25 kPa of stress. Beyond this threshold, tremor sensitivity to additional stress changes decreases, even as aseismic slip continues[19]. The subsequent

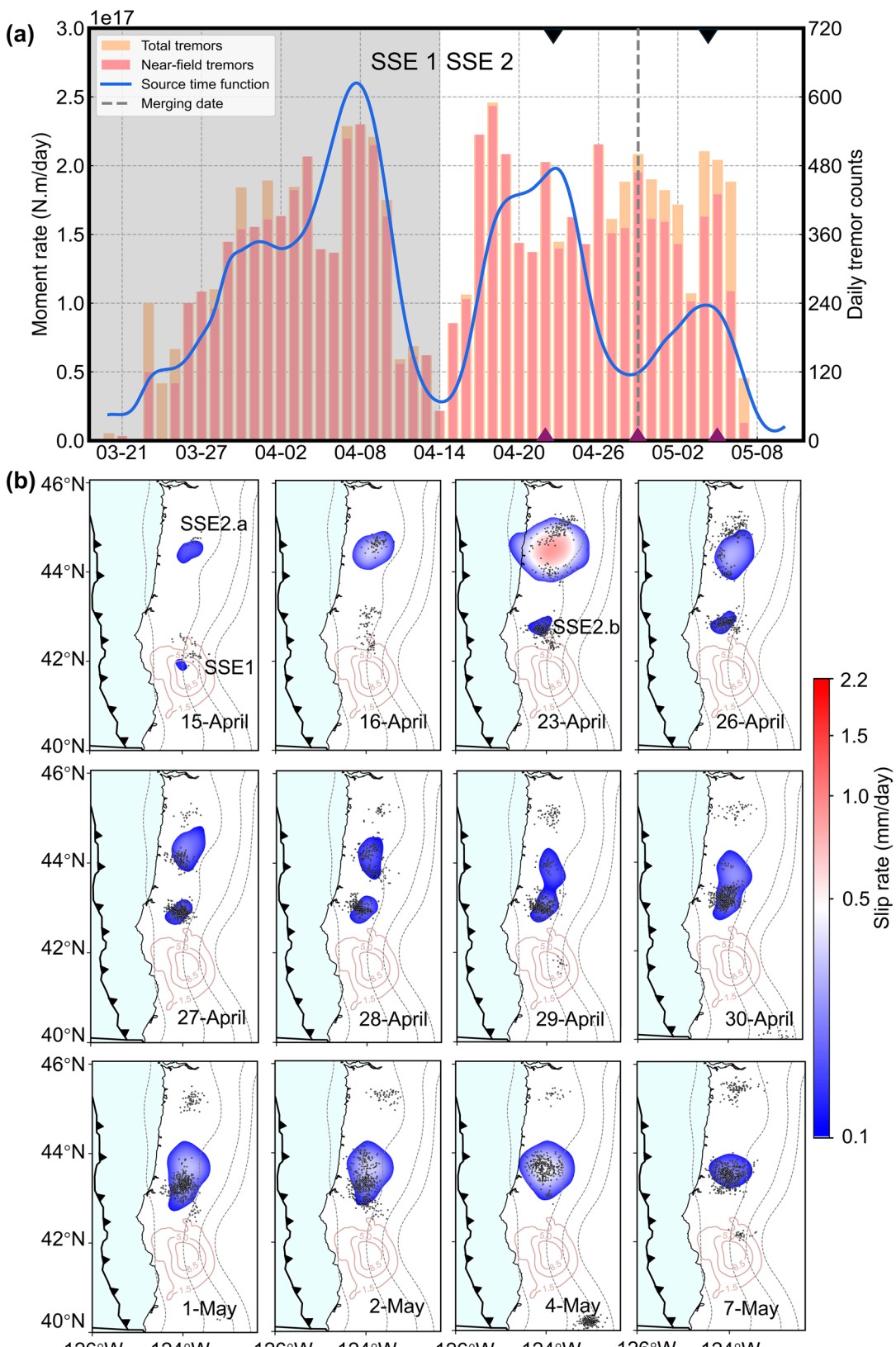

**Fig. 1 | The moment rate and statistics of episodic tremor and slip (ETS) in 2021 and the evolution of slow slip event (SSE). a** Moment rate history (blue curve), and daily near-field (within 50 km) and total number of tremors (red and yellow histograms, respectively) during the whole ETS. The vertical dash lines indicate the coalescence time of SSEs. The gray shaded area is used to distinguish the two SSEs. The two black triangles at the top of the x-axis indicate the two peak values of the moment for the SSE2. The three purple triangles at the down of the x-axis indicate the three increases of the tremors for the SSE2. **b** Spatio-temporal evolution of SSE2. Daily slip rate contoured in color maps. The orange contour lines indicate the cumulative slip area every 3.5 mm of SSE1. The black dots represent the corresponding daily tremor locations from the Pacific Northwest Seismic Network (PNSN) catalog (https://pnsn.org). The black dashed lines indicate depths at 20 km interval[40].

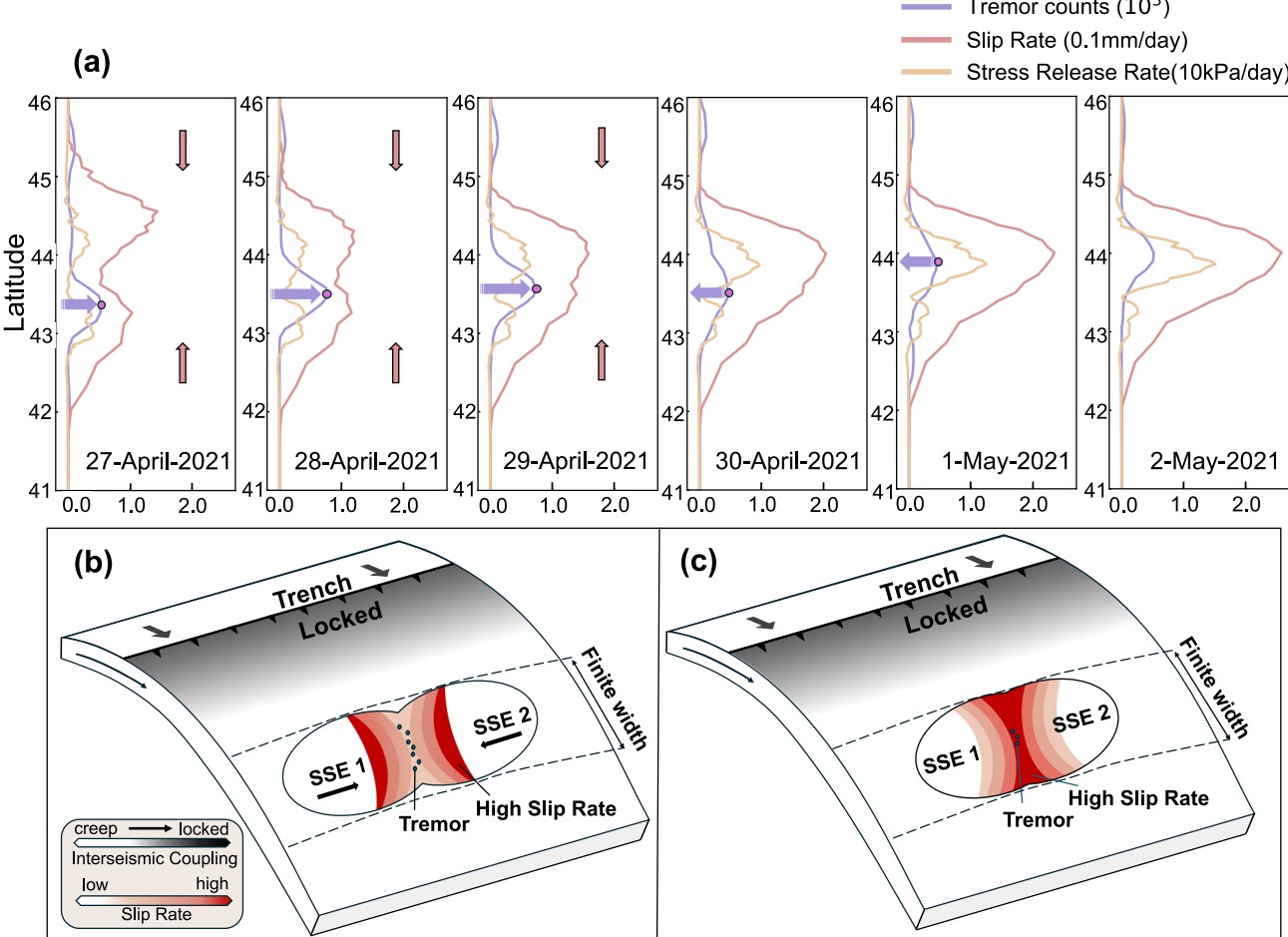

**Fig. 2 | Stress and tremor count variation during coalescence of slow slip event (SSE) in the deceleration phase. a** Stress changes from April 27th to May 2nd. The purple line represents the number of tremors, the red line represents the slip rate, and the yellow line represents the stress release rate along the strike. The red arrows indicate the direction of slow slip fronts propagation, and the purple arrows represent the increase or decrease in tremor counts in the coalescing area. **b** Schematic diagrams of tremor and slip in the early stage of the coalescence of two SSEs. **c** Schematic diagrams of tremor and slip in the intermediate stage of two SSEs coalescing.

increase in tremor counts occurs at latitude 43.5°N. At the same location, tremor counts continues to decrease over the following days, while the moderate tremor peak migrated to the north at latitude 44°N.

**Simulation of SSE coalescence and influencing factors**

The deep learning-based SSE catalog and GNSS data inversions have revealed SSE coalescence and the resulting slip acceleration. Aside from the 2021 coalescence event, a similar coalescing event during the deceleration period also occurred at the same location in 2019, with the moment rate showing a bimodal characteristic (see Figs. S22 and S56). This implies that regional physical parameters may influence or control the repeated occurrence of such coalescing events. Next, we investigate the factors that drove coalescence during the deceleration period. Numerous observational[20–22] and simulation studies[23,24] indicate that heterogeneous features, such as frictional parameters and stress state, play a critical role in controlling SSE behavior. For instance, Michel, et al.[25] proposed an along-strike segmentation of the CSZ into 13 distinct segments based on the extent of their detected SSEs from GNSS data, while Li and Liu[26] segmented the CSZ into 5 segments based on gravity anomalies, which they used to infer a distribution of effective normal stress parameters for numerical simulation of SSEs. Here we developed rate-and-state friction models to examine SSE coalescence and its controlling factors (Fig. 3), using

quasi-dynamic (QDYN) simulations[27] with a modified rate-and-state friction (RSF) law (see "Methods"). The model is configured to mimic Cascadia-like SSEs, characterized by an elongated aspect ratio and slip that saturates the width of the SSE zone[4,10,28,29]. We consider three heterogeneous models to investigate coalescing events during the deceleration phase: (Model I) a model with a-b variations along the strike; (Model II) a model with characteristic slip distance (Dc) variations along the strike; and (Model III) a model with variations in effective normal stress ($\sigma_n$) along the strike. Further details on the model setup are provided in Methods.

The recurrence interval of SSEs in all three models is ~1.5 years, which aligns with actual observations (Fig. S17). The simulated SSEs tend to fill the entire depth range of the VW zone and typically propagate at a rate of several kilometers per day. All three models can simulate SSE coalescence scenarios (see Fig. 3), indicating that SSE coalescence is a complex phenomenon influenced by multiple factors. The moment rate exhibits a bimodal pattern, with a secondary stress peak caused by slip front coalescence.

First, a homogeneous model in terms of a − b, Dc and $\sigma_n$ also generates coalescing SSEs. This reference model suggests that coalescing events does not require complex model setups (Fig. S18). In model I, variations in the a − b parameter add complexity to the model and create contrasts in the propagation speed of events, which can in turn increase or decrease the occurrence of such events (Figs. S19, 20).

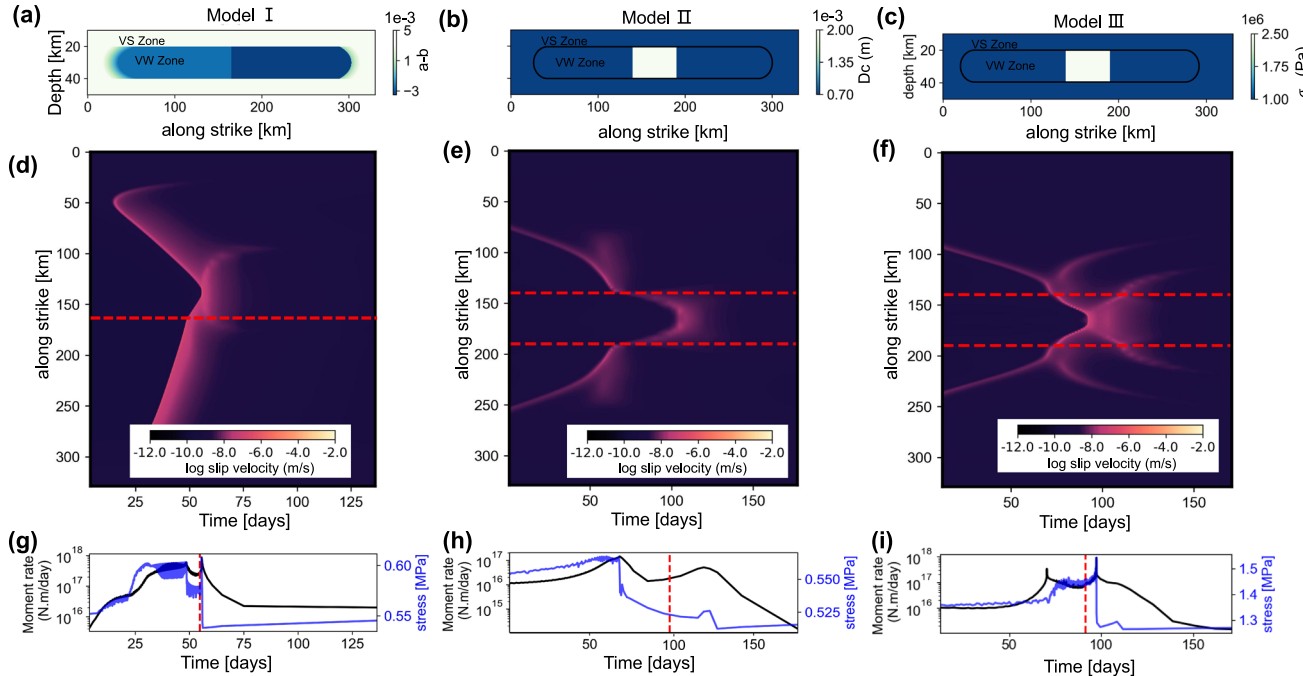

**Fig. 3 | SSE numerical simulation. a–c** Three model configurations, each comprising VS (a-b > 0) region surrounding a central VW (a-b < 0) patch. **a,** In the model I, the VW region exhibits variations in a-b along the strike. **b,** In the model II, the VW region shows variations in Dc along the strike. **c,** In the model III, the VW region displays variations in $\sigma_n$ along the strike. **d–f,** Spatiotemporal evolution of slip rates following the coalescence of SSEs during the deceleration phase for the three models. In panel d, the red line represents the heterogeneous variation of a-b. The horizontal red lines in panels b and c display the edges of the barrier. **g–i,** Temporal evolution of moment rate and maximum stress for the coalesced events in the three models. Due to the resolution of the patches setting, the measured maximum stress exhibits oscillations. The red dashed line indicates the time of slip front coalescence.

Slowing propagation speed and increasing event duration make it more likely for another SSE to occur and collide with ongoing event. Similarly, in Model II and III, the bounded zones of large contrast in Dc and $\sigma_n$ (1.5 MPa and 1.3 mm, respectively) also slowing down the propagation of SSEs, making those zones more likely locations for coalescence. Nevertheless, as $\Delta Dc$ and $\Delta\sigma_n$ increase, they might act as an actual barrier, thus preventing or reducing SSE coalescence).

We now compare the results of the three models with the actual observations of the 2021 coalescing SSE. In Model I, the moment rate peaks almost immediately after SSE fronts coalesce and decays rapidly, consistent with the rapid release of instantaneous slip rate during the acceleration phase coalescence observed by Bletery and Nocquet[9]. In contrast, Models II and III display a notable time delay of the moment rate peak after the coalescence. Model II reaches its peak ~18 days after coalescence, while Model III attains a secondary peak around 6 days, aligning with the observed moment rate peak on May 4 following the SSE coalescence on April 28, 2021. Note that the actual values of Dc and $\sigma_n$ might control the delay between the start of the coalescence and the moment rate peak. Regarding rupture propagation velocity, Model I concurs with observations, exhibiting distinct migration speeds for the two SSEs (Fig. S23). As mentioned, Models II and III show secondary variations in migration velocity due to barrier effects (differences in Dc or $\sigma_n$). This is consistent with the observed slowdown along-strike tremor propagation velocity as tremors approach the coalescing zone (Fig. S23). Note that the 2021 SSE occurred around 44°N, where the northern part lies beneath the central Oregon segment with low tremor density, associated with Siletzia, the accreted oceanic basalt terrane composing the Oregon forearc[30,31]. The southern part corresponds to the Klamath Mountains, where the deep structure involves lower velocity, quartz-rich sediments of the Franciscan accretionary complex[32]. These geological differences can potentially result in variations in frictional properties and critical slip distance between the two regions. Wells, et al.[33] suggested that the fault in this region extends to an overpressured megathrust, providing fracture channels for fluid escape into the upper plate, potentially reducing fluid pressure at the location of SSEs and resulting in heterogeneous effective normal stress. We propose that the 2021 coalescence event could have resulted from a combination of those factors, but further analysis is needed.

## The scaling law for SSEs

A key feature of SSEs coalescence is the spatial convergence of multiple SSEs. Unlike coalescence during the acceleration phase, coalescence during the deceleration phase requires at least one SSE to migrate, owing to the natural decline in slip rate and rupture area (see Fig. S1). We categorize SSEs in the catalog into migrating SSEs and non-migrating SSEs by examining whether centroid of SSE ruptures and tremors migrate along the strike. Coalescence of SSEs in the deceleration phase is considered a special case of migrating SSEs. To investigate the properties of coalescing SSEs, we analyzed the scaling relationships governing SSEs with different migration behaviors[25]. The analysis includes a total of 49 SSEs (Fig. 4a, b). By comparing the moment-duration relationships of SSEs with previously established scaling laws[25,34–36]:

$$\log_{10}(T) = \left(\frac{1}{c}\right)\log_{10}(M_0) + g \tag{1}$$

where $T$ is the duration (in units of seconds), $M_0$ is the event moment (in units of N m) defined as the integral of slip over the rupture area multiplied by the shear modulus, and $c$ and $g$ represent the slope and intercept, respectively. The optimal scaling exponent c = 2.8 falls between previous values, where Ide, et al.[34] found c = 1 and Michel, et al.[28] suggested c = 3 (Fig. 4c). In any case, migrating SSEs release more moment[37] compared to non-migrating SSEs (red box in Fig. 4c). We then fit the c values for migrating SSEs and non-migrating SSEs

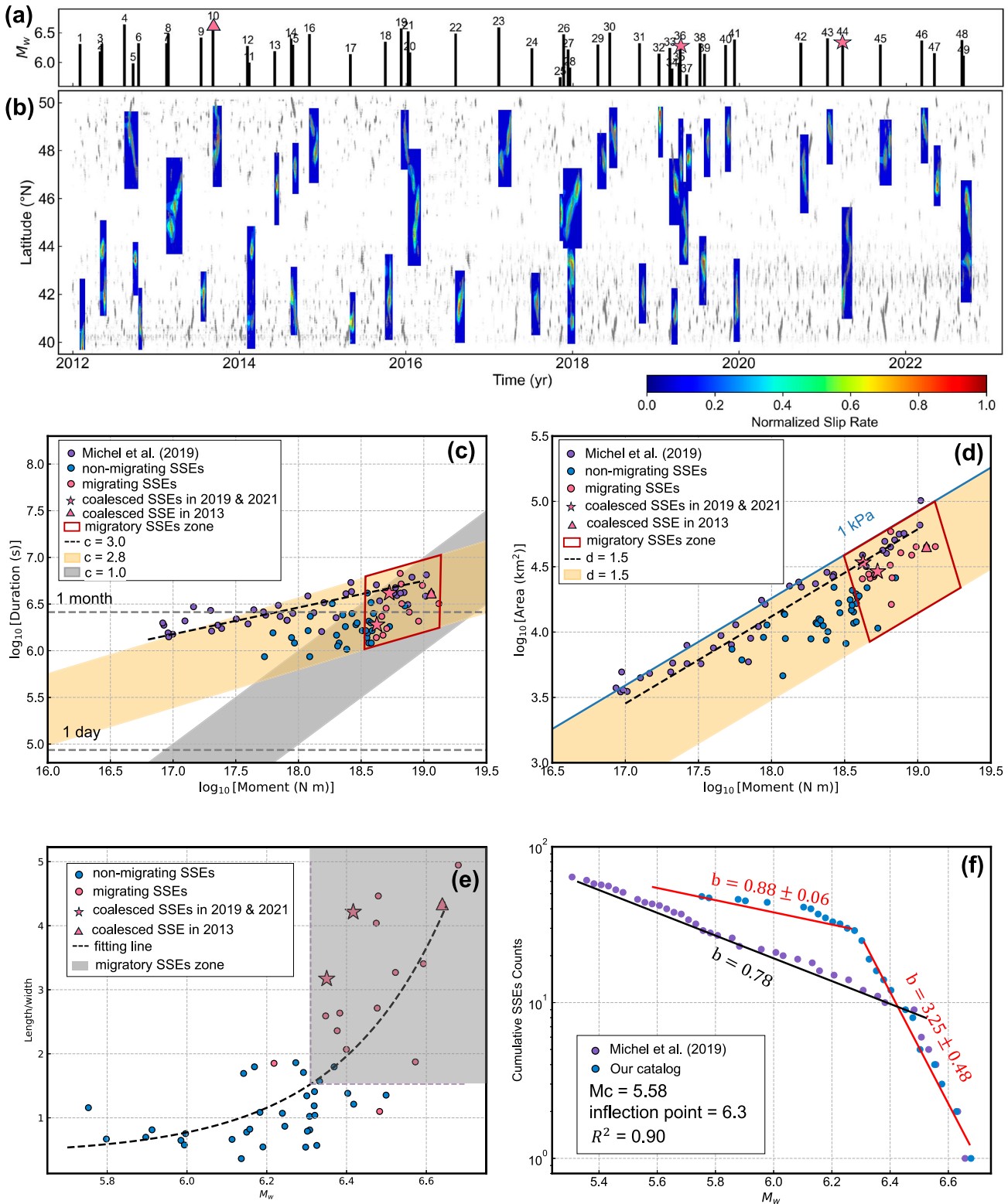

**Fig. 4 | Spatio-temporal distribution of slow slip events (SSEs) and the scaling laws of SSEs. a** Time and magnitude distribution of SSEs in our catalogue. **b** Timing and rupture extent of the SSEs. The gray dots represent the corresponding daily tremor locations from the PNSN catalog. The slip rates have been normalized for plotting. **c** Relationship between moment and duration of SSEs. **d** Relationship between moment and rupture area of SSEs. The blue line represents the contour of 1 kPa stress drop. **e** Aspect ratio of rupture areas. **f** The magnitude-frequency distribution. Purple circles indicate Michel's catalog's magnitude-frequency distribution, and blue circles show our catalog's distribution.

individually. The results show that the $c$ value for migrating SSEs is 1.5, while that for non-migrating SSEs is 3.6. Gomberg, et al.[36] found that when the rupture reaches slip zone boundary and the 2-D rupture growth transitions 1-D, the $c$ value changes from 3 to 1. The difference

in the $c$ values between migrating and non-migrating SSEs can be attributed to the finite-width model. No clear trend is seen for coalescing events. One might expect these types of events to exhibit a larger moment release than their non-coalescing counterpart due to

their additional burst in slip rate, however, it does not lead to a change in the magnitude level. Additionally, the duration of coalescing events might be shorter than expected for their magnitude, as multiple areas are slipping simultaneously rather than being ruptured more gradually as for non-coalescing events.

In addition, we explored the scaling relationship between moment and rupture area, observing that SSEs follow a trend roughly consistent with the scaling $M0 \propto A^{(3/2)}$ [25], akin to regular earthquakes (Fig. 4d). While our best-fit scaling exponent aligns with previous findings, the intercept is lower, potentially due to variations in model regularization methods[28]. Furthermore, migrating SSEs show elongated rupture areas (Fig. 4e). Coalescing events might yield further expansion of the slip area along strike (Fig. 4d). However, assuming constant SSE width, coalescing events during the deceleration phase seem to be longer than the non-coalescing SSEs (Fig. 4e). Further observations and analysis are needed to confirm such results.

The frequency-magnitude distribution of SSEs reveals distinct inflection points, suggesting unique scaling behaviour at different magnitudes (Fig. 4d). For instance, earlier studies[25] observed a *b* value of 0.78, with a sudden decrease in event frequency beyond Mw 6.4. Our findings indicate a similar distribution, but with a more pronounced inflection, yielding a b-value of 0.88 for Mw < 6.3 and a steep increase to 3.25 for Mw > 6.3. We noticed that the SSE aspect ratios also exhibit a similar inflection point (Mw=6.3) when using the evaluation by a Dynamic Programming approach (DynP)[38] (See "Methods"). By fitting the aspect ratio data to an exponential function, the corresponding aspect ratio is found to be 1.65 (See "Methods"). Regions with an aspect ratio ≥1.65 and Mw ≥6.3 were strongly associated with migratory behaviour (Fig. 4e), suggesting that inflection in SSE frequency-magnitude scaling result primarily from along-strike migration and complex slip behaviours, including coalescing and splitting.

## Discussion

Our study shows that SSEs can coalesce not only during the acceleration phase but also during the deceleration phase. The 2021 event demonstrated that the coalescence of the slip fronts during the deceleration phase led to a secondary acceleration of slip velocity and the expansion of the rupture area, forming a secondary peak in the moment release rate. This phenomenon reveals the complexity of SSE interactions, where the coalescence counteracts the slip attenuation during the deceleration phase. Numerical simulations suggest that the 2021 event might have been a coincidental occurrence favored by regional stress heterogeneity. Coalescence during the deceleration phase requires at least one SSE to migrate along the strike. Scaling law analysis indicates that migrating SSEs release more moment, involve larger rupture areas, and exhibit a frequency-magnitude distribution with a turning point at Mw 6.3. Coalescing SSEs can be considered a specific case of migrating SSEs. Although it is difficult to extract specific trends in the scaling laws for these events due to their small number, the ones occurring in the deceleration phase seem to be potentially longer than the other migrating events of similar magnitude.

Past studies have often divided the CSZ into different segments based on cumulative slip distributions, tremor distributions, and gravity anomalies[25,26]. Our research highlights that SSEs and tremors may overcome these segment boundaries due to factors such as stress memory and that SSEs in different regions can coalesce. However, the coalescing characteristics of SSEs vary across regions. The SSE coalescing events in the northern CSZ (2013 events) occur during the acceleration phase, leading to slip bursts phenomena. In contrast, the SSE coalescing events in the southern CSZ (2019 and 2021 events) occur during the decelerating phase, showing clear migration behavior, that results in a secondary acceleration of the moment rate. Both coalescence during the acceleration and deceleration phases lead to

an increase in slip rate. However, the impact of SSE coalescence on the spatial and temporal range of future large earthquakes remains unclear. We speculate that the acceleration of slip rate and expansion of rupture area caused by SSE coalescence could significantly alter the redistribution of fault stress, influencing the occurrence or rupture process of future earthquakes. Future work will need to use more refined numerical simulations and long-term observations to quantify the contribution of coalescing SSEs to stress fields and earthquake cycles, assessing their role in earthquake disaster forecasting. In conclusion, SSE coalescence during the deceleration phase highlights the role of fault heterogeneity and migration behavior in the dynamics of SSEs.

## Methods

### SSEs detection using machine learning

We adopt daily position timeseries processed and combined by the Scripps Orbit and Permanent Array Center (SOPAC) and the Jet Propulsion Laboratory (JPL) from the Network of the Americas operated by EarthScope Consortium. We devise an approach[12] that first employs the variational Bayesian Independent Component Analysis (vbICA)[39] to improve the signal-to-noise ratio of GNSS time series (Fig. S4) and then utilizes deep learning combining bidirectional Long Short-Term Memory, the Channel Attention Module and the Spatial Attention Module to identify SSEs (Text S1, 2). We apply this method to the GNSS three-component time series at 240 stations along the Cascadia subduction zone (CSZ) from 2012 to 2022 (The location of the stations is shown in Fig. S2). After inspecting the inversion results, we have retained 49 SSEs detected by the deep learning model (The detection results of the machine learning are shown in Figs. S5, 6).

### Kinematic slip model

The Cascadia subduction interface is discretized into 802 quasi-equilateral triangular sub-faults with 19.6 km long edges, following the curved surface from the Slab2.0 model[40]. The fault area extends from latitude 39°N to 51°N and from the trench down to 90 km depth (The fault geometry is shown in Fig. S3).

We simultaneously solve for the spatially variable daily slip rate using the Green function method.

$$\sum_{t_k=0}^{t_k < t} Gm(t) = d(t) \qquad (2)$$

where $G$ is the Green's function, calculated in a homogeneous elastic half-space for triangular dislocation[41]. $m(t)$ is the constant slip rate for all dates and $d(t)$ is the time series vector. The rake is calculated on each sub-fault accounting for the rotation of the North American plate to the Juan de Fuca plate[42]. The inverse problem is highly underdetermined. We add regularization constraints in the form of a minimum Laplacian in space and time. This method permits different spatial and temporal smoothing constraints, governed by the dimensionless constants $\lambda_{space}$ and $\lambda_{time}$, respectively[43]. We conducted an L-curve test for SSE in 2021 to select the optimal model, which has regularization parameters $\lambda_{space} = 4$ and $\lambda_{time} = 1$ (Fig. S8), and we performed inversions on all SSEs using the same parameters. The results of the dynamic resolution test (Text S3) are shown in Figs. S12, 13. The corresponding wave fits are available in Figs. S14–17. The moment rate release (Fig. 1a) is calculated by summing the contributions of all sub-faults assuming a uniform elastic Earth with an elastic shear modulus of 30 GPa. The complete set of inversion results is presented in Supplementary Figs. S26–68 and Supplementary Movie 1.

### Numerical methodology for simulating SSEs

We employ the QDYN software, a boundary element method that simulates SSEs under a quasi-dynamic approximation within a rate-

and-state friction framework[27]. We adopt an updated rate-and-state friction law with a transition from VW to VS at steady-state with increasing slip rate [4,44–47]:

$$\frac{\tau}{\sigma_n} = \mu^* - a\,ln\left(\frac{V_1}{V}+1\right) + b\,ln\left(\frac{V_2\theta}{D_c}+1\right) \qquad (3)$$

$$\frac{d\theta}{dt} = 1 - \frac{V\theta}{D_c} \qquad (4)$$

where $\tau$ and $\sigma_n$ are the shear stress (frictional strength) and effective normal stress (normal stress minus pore fluid pressure). Respectively, $V$ is the sliding velocity and $\theta$ is a state variable, $\mu^*$ is the reference friction coefficient, $D_c$ is the characteristic slip distance of state evolution, a and b are friction parameters controlling the velocity and state effects, respectively. The difference (a–b) defines the frictional regime on the fault. If a–b > 0, the steady-state friction is VS; If a–b < 0, steady-state friction is VW. Cut-off velocities $V_1$ and $V_2$ account for both direct and indirect effects such that fault material can transition from VW to VS at a higher slip rate (with a–b < 0). The model setup and parameters are provided in Supplementary Text S4 and Table 1.

## Scaling law

We compare the best-fitting moment–duration scaling law of our catalog with Michel, et al.[25]:

$$\log_{10}(T) = \left(\frac{1}{c}\right)\log_{10}(M_0) + g \qquad (5)$$

where $M_0$ is expressed in $N\,m$ and $T$ in seconds. We used least squares regression to calculate the parameters. The best fit corresponds to exponent $c = 2.8$ in our catalog.

We use a similar procedure to calculate the best-fitting moment–area scaling law:

$$\log_{10}(A) = \left(\frac{1}{d}\right)\log_{10}(M_0) + r \qquad (6)$$

where $M_0$ is expressed in $N\,m$ and $A$ is the cumulative slip area in $km^2$. The best fit corresponds to exponent $d = 1.5$, which is consistent with Michel's result.

We use the Dynamic Programming approach (DynP)[38] to determine the inflection point of aspect ratios data. DynP recursively divides the data and calculates the cost of different divisions to find the optimal inflection point. The recursive formula is as follows:

$$D[k,j] = \min_{i=1}^{j-1}(D[k-1,i] + cost(X_{i,j})) \qquad (7)$$

where $D[k,j]$ represents the minimum cost for selecting k inflection points (here, k = 1) from the first j data points. $cost(X_{i,j})$ denotes the cost of the data $X$ from point i to point j, and we choose the L2 cost:

$$cost(X_{i,j}) = \frac{1}{j-i+1}\sum_{k=i}^{j}(x_k - \bar{x}_{ij})^2 \qquad (8)$$

where $X_{i,j}$ is the data segment from index i to j, $\bar{x}_{ij}$ is the mean of that segment. We then fit the aspect ratio data to an exponential function to find the aspect ratios corresponding to the inflection points:

$$y = y_0 + A_1 e^{-(x-x_0)/t_1} + A_2 e^{-(x-x_0)/t_2} \qquad (9)$$

The final curve obtained is $y = 0.42116 + 0.00372e^{((x-4.99788)/0.27039)} + 0.0049e^{((x-4.99788)/0.27029)}$.

We calculate the magnitude-frequency distribution by maximum likelihood method[48–50]:

$$\log_{10}N = a - bM \qquad (10)$$

where $N$ is the number of SSEs with magnitude larger or equal to $M$, b is a scaling parameter and a is a constant. More details are provided in Supplementary Text S5 and Figs. S23.

## Data availability

The Cascadia time series are available at (ftp://garner.ucsd.edu/pub/timeseries/measures/ats/WesternNorthAmerica), and the tremor data is available at PNSN websites (https://tremor.pnsn.org).

## Code availability

The kinematic slip inversion code [PYEQ52] is available on Zenodo at (https://zenodo.org/record/7467256#.Y6Ljb7KZNb8). The rate-and-state earthquake simulator (QDYN)[27] used in this work is available at (https://github.com/ydluo/qdyn). The deep learning-based SSE detector[53] is available at (https://github.com/evedor/SSE_Detecter).

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

## Acknowledgements

K.C. was funded by National Natural Science Foundation of China (Nos. 42474046 and 42074024). L.D.Z. was supported by the Earth Observatory of Singapore (EOS), the Singapore Ministry of Education Tier 3b project "Investigating Volcano and Earthquake Science and Technology (InVEST)" (Award No. MOE-MOET32021-0002) and the Nanyang Assistant Professorship (NAP) (Award No. 025244-00001). S.M. was supported by the French government through the UCAJEDI Investments in the Future project (ANR-15-IDEX-01) managed by the National Research Agency (ANR). S.H. was funded by The Key Research and Development Program Project of Jiangxi Province (No. 20243BBI91033). We are grateful to Jean-Philippe Avouac for his careful review and valuable suggestions on an earlier version of this manuscript, which significantly improved the manuscript. Some of the figures were plotted by Generic Mapping Tools (version 6.0.0)[51].

## Author contributions

K.C. conceived and led the study. W.J. conceived and designed the study, conducted the experiments, and wrote the initial draft. S.M. and L.D.Z. wrote the paper and contributed to the interpretation of the results. H.Z., L.X. and S.H. assisted with the processing of GNSS data and finite element inversion. J.X. assisted with numerical simulations and contributed to the interpretation of the results. All authors contributed to the final completion of the manuscript.

## Competing interests

The authors declare no competing interests.
