## [Transparent Peer Review file · Nature Communications]

Secondary acceleration of slip fronts driven by slow slip event coalescence in subduction zones

Corresponding Author: Professor Kejie Chen

Version 0:

Reviewer comments:

Reviewer #1

(Remarks to the Author)

Wang et al. present a study on secondary acceleration of slip fronts associated with the coalescence of slow slip events (SSEs) in Cascadia. They perform kinematic slip inversions of GNSS data for all events that have happened between 2012 and 2023. They find a coalescence episode of SSEs in 2021, adding up to the one previously identified in 2013. Confirming the findings of Bletery and Nocquet (2020) for the 2013 case, they find that this merging episode is associated with slip acceleration. They further present 3 numerical models explaining this behavior. Additionally, they discuss the specificity of migrating SSEs based on scaling laws.

Overall, I find the study interesting. This second observation of an SSE coalescence deserves to be published. Nevertheless, I do have a list of comments / questions that I hope will help improve the manuscript.

On the structure of the article:

The final part of the manuscript (on scaling laws of migrating SSEs) appears a bit disconnected from the rest. Consider clarifying the connection to the rest of the paper.

More generally, the manuscript is a bit "messy" with 65 supplementary figures appearing in a rather chaotic order, passed comments appearing in the middle of sentences (see minor comments) and references appearing multiple times (e.g., ref 10 and 15).

On the inversion of the coalescence episode:

I see in Fig S7 that only about a third of GNSS stations have been used in the inversion. Why?

I would strongly recommend performing the inversion on all available stations. Would that change the results?

On the question of "merging vs normal" SSEs:

Define what a "normal" SSE is.

I am guessing that you mean a non-migrating SSE. This brings the question of the resolution of the kinematic inversions. Migrations are much easier to see for large SSEs than for small ones. This could explain why migrating SSEs appear to have larger moment (and therefore larger extent and duration) than "normal" ones. One way to investigate whether "normal" SSEs truly do not migrate would be to look at the associated tremor distributions. Consider strengthening the discussion on SSE migrations based on tremor migrations.

Minor comments:

L21-22: I think the sentence " can influence the timing and 21 spatial extent of future seismic activity in the region. " was added by mistake in the middle of another sentence.

L29: List factors.

Fig 3a-c: What do the colors correspond to? Contrasts in D_c and Σ_n ? Explain in the caption.

Reviewer #2

(Remarks to the Author)

Dear editor and authors,

I thank you for the opportunity of reviewing this work.

In this paper, the authors study a catalogue of slow slip events (SSE) in the Cascadia subduction zone. The catalogue was obtained by analyzing continuous GNSS recording lasting 14 years. They also compare these data to a catalogue of tremors. 49 SSE are detected. For each, the slip rate evolution on the fault zone is inverted., and the stress change evolution is estimated. The authors investigate the special case of an SSE coalescence. They describe the spatio-temporal evolution of this SSE., and its relation with the tremors evolution. Moreover, they propose dynamic simulation base on a rate and state law.

The data, and data analysis, providing the catalogue, of SSE are really interesting. Moreover, the detailed study of the coalescence event is also of interest. However, the paper suffer from a lack of logical structure, and the findings are not satisfactorily discussed and analyzed altogether to drive clear conclusions or interpretations. Moreover, a number of typos, and editing problem in the figures must be corrected. To conclude, I find that the scientific content is solid and valuable, but that the paper needs major revision in order to convey a clear message.

Here are some of the major points that impeded the understanding of the paper and need to be addressed

1. It is said L84 that the catalogue is available. But then in the methods and sup mat, it is presented as if it is the core of this work to build this catalogue. So this is unclear and should be clarified. Then, if this is the core of this work, the catalogue should be presented at the beginning, and not at the end.
2. It is not clearly explained how SSE are classified as “normal”, “migrating” or “coalescing”.
3. Overall, there seems to be some confusion between slip acceleration (an increase of slip rate) and front acceleration (an increase of migrating velocity of the fronts). Both can be linked, but when the two events have merged, there isn't anymore fronts. So it is not clear what is the message the authors which to convey: are the slip front accelerating before they merge? or is the slip rate increased upon coalescence? Maybe this confusion arise from the fact that what is called a slip front is not really described? Is it the limit of the slipping zone? or the area in which the slip rate is larger? How are these two related?
4. Why do the authors chose to focus their analysis on this specific coalescence event? Is it the only one observed besides the 2013 event? Would it be worth to conduct the same analysis on the 2013 event, or too redundant with previous studies. Are these the only 2 coalescing events in the catalogue?
5. It is absolutely not clear why the numerical model were conducted, why heterogeneity was introduced, and most of all, how they compare to the data and/or help to understand the observations. THeY look like they are interesting by themselves but are not discussed.
6. A global, analytical discussion of all the findings is needed, as well as a clear conclusion.
7. A lot of figure in the supplementary material cannot be read entirely because the lack the resolution to do so even when zooming on a screen computer.
8. The detailed methods can stay in the Methods section, but a clear simple summary of the work conducted is needed in the main text, at the beginning. As is, the reader cannot understand how the slip rate and stress changes curves come from.

Here are some minor typo or problems that should also be addressed or corrected..

- L21-22: there is a piece of sentence that should not be here
- L24. What is called a proto SSE?
- L29 can you state the factors, or the most important ones?
- L34: “which can modify stress”: remove can, it does modify stress.
- figS1 B: The sketch is not really clear to me, because if there is merging, there should be 2 SSE depicted. S1b: accelerate faster is not clear. do you mean higher acceleration, do you mean (accelerate more=higher acceleration) or (time derivative of the acceleration grows) .
- L51: word or end of sentence missing.
- L84: is the catalogue available, or have you built it as explained in sup.mat.
- test S2 L56 60: is this output probability of SSE or occurrence and duration of SSE.?

- fig S3: color scale cannot be read, improve resolution or make it bigger.
- fig S4: SSE 22 appears two times SSE 23 is missing.
- Fig 1: What is the difference between a near field tremor and a total tremor. Why are they not discussed in text? Why do tremors remain active in the north while no slow-slip is detected?
- Fig 2: text and label a little bit too small. There are editing problems in fig 2a (grey boxes around label)
- L138-140: I find that the description does not correspond to the figure. The tremor activity looks almost constant during the whole second part, and I do not think that the peak during the lower slip rate is minor, but rather larger.
- L141. I do not understand this interpretation. From the data it looks like the tremors are located inside the high slip rate area, so why do you state that they occur in a locked area?
- L147-151: what is the analysis that gives you the shear stress rate, where is it presented?
- L153-166: The description is unclear
- L164: I did not understand what is the physical value corresponding to 25kPa?
- L177-133: see main remarks number 5
- L215-249. At this point, it is unclear if you are describing simulation results or the actual catalogue. See main remarks number 1. The results are presented but not interpreted
- L220: define T_c g M0
- Figure 4.4a: improve resolution, what are grey dots? why are certain parts in color,...? 4f include legend in the figure.

I hope that this review comes out helpful for the authors and I thank again the editor and the authors for the opportunity of reviewing this work.

Reviewer #3

(Remarks to the Author)

This paper focuses on the analysis of coalescence SSEs in the Cascadia subduction zone. It performs a detailed analysis of one event in 2021, extracted from previously published SSE catalog, and also numerical simulations to estimate the impact of various frictional properties on SSE characteristics, and a discussion on SSE scaling laws.

While there are several interesting results and discussion in the paper, the link between the different aspects discussed is not clear, and as a result I find the messages of the paper difficult to follow.

I detail below my main comments regarding the 3 main points discussed in that paper:

The first one, related to the detailed analysis of the 2021 coalesced ETS event, is well detailed, and brings interesting observational constraints on the relationships between tremors and slip, and on the moment rate evolution when two SSEs merge. This study is rather clear and well detailed.

The second point related to the rate and state modeling of SSE cycles and factors controlling the coalescence: the results presented are interesting, and different behaviors are observed depending on the assumed frictional properties. However, the paper only provides a rather superficial description of the results of the different models, but does not provide any clear interpretation, nor relate the models to the observed coalesced SSE in Cascadia. I would have appreciated a discussion on which model (I, II, III) better explained the observed ETS sequences in Cascadia. Which moment rate functions are more similar to the observations? The models also seem to exhibit very different amounts of slip per event and various rupture velocities (from fig. S18), these points are not at all discussed in the paper. It is thus difficult to assess how the numerical simulation helps to understand the coalescence process.

The third point discussed relates to the scaling laws for SSEs: once again, I find it difficult to relate this point with the previous ones. There are only two examples of coalesced SSEs in the catalog (they are among the largest ETS observed), and the discussion here is more focused on the transition between migrating / non-migrating SSEs, and it is not clear to me what it brings to the understanding of the coalesced SSEs.

The scaling figure (Fig.4) also brings several questions, not discussed in the paper, and listed below:

- Some of the events in the current study are also present in Michel et al.'s catalog: did the author make a systematic comparison of moment and duration from both catalogs for similar events? Are the results consistent?
- what are the three more small events in Michel et al.'s catalog's with respect to the current one?
- what is the origin of the difference between the scaling from Michel et al.'s catalog and the current scaling (in particular, why did they not observe an inflection point?).

-I also did not get the criteria to consider a SSE as “normal” or “migratory”.

While there are several interesting results and discussion in the paper the link between the different aspects is not clear, points 2 and 3 lack some details, it is not clear what they bring to the understanding on the coalesced SSEs. As a result, I find the message(s) of the paper difficult to follow.

At its current stage, the paper gives the impression of separate studies being combined into a single manuscript without a clear unifying message. If points 2 and 3 are intended to contribute to a broader discussion on coalesced SSEs, this should be made more explicit. Otherwise, the authors might consider dividing the manuscript to present a clearer and more focused message for each component.

In my opinion, the structure and central message of the paper needs to be clarified before it can be considered for publication.

Minor comments:

l.21 “evoluti” => evolution

l.22 “region. on of slip” => check sentence

l.51 “elevating seismic”: missing word ? please clarify

l.71-72 “and is different from the 2013 event”=> the 2023 event (ref. 15 Bletery & Nocquet) has not been discussed yet so the comparison is not very relevant. Maybe discuss or provide a reference for this event earlier.

Figure 2a: it is unclear what are the units for each plots. Are they normalized ?

In supplements:

l. 37: “we employ three times of vblCA” => check grammar.

Version 1:

Reviewer comments:

Reviewer #1

(Remarks to the Author)

I am satisfied with the revisions made by the authors.

I recommend publication pending minor revisions (listed below) that I believe the authors can do without going through an additional round of review.

Minor comments:

_ The quality of writing is good enough for someone familiar with the topic to follow but might be an obstacle for a broader audience. Re-writing with the help of a native english speaker would help broadening the impact of the paper.

_ The 2019 event could be discussed a bit more in the main text.

_ From Fig 4c, I get the impression that migrating and non-migrating SSEs might follow different scaling laws (with $c = 1$ and $c = 3$, respectively). I would recommend to fit c values for both populations individually and to discuss the results (if correct this would explain the question of the scaling laws by the saturation of the width of the transition zone).

I hope this helps.

Reviewer #2

(Remarks to the Author)

please see the review in the pdf file attached.

[Editorial Note: The PDF is appended to the end of this file]

Reviewer #3

(Remarks to the Author)

The authors have significantly clarified the message of the paper and the connections between the different parts. I still have a few remarks and comments that could be clarified before publication.

New figure S1 and discussion on coalescing events

I do not completely agree with the features and discussion presented in the new additional figure S1. The authors claim that the coalescence of SSE in the deceleration phase requires at least one migrating event.

It seems to me that all SSEs that reach a certain size (when they fill the width of the SSE zone) start to migrate. This is what is shown in Fig. 4, where all large – and only large SSEs are migrating. I thus do not fully agree with the

sketch presented on Figure S1:

The different in size between migrating / non migrating SSE is not shown here, and the aspect ratio between the circular growth versus along strike propagation are not clearly represented. While it is not required for a sketch to be at scale, I think this lack of scaling is misleading, as some on the scenari presented might not be detectable from a geodetic point of view: I'm not sure that the case "a" where two SSEs are merging during the acceleration phase without migration would be detectable from a geodetic point of view: for two non-migrating SSEs to coalesce, this would require their initiation to be very close (less that the width of the SSE zone, assuming that the SSE initially grow as a circle. Do you think geodetic inversion which include some spatial smoothing, would be able to robustly separate initiation points that are very close ? It is misleading to represent the same distance between initiation points in case "a" and "c" in Fig. S1, as the distance would need to be much closer in case "a" than in case "c" for the events to coalesce.

Do you have examples of non-migrating coalescing SSEs ? Unless I am mistaken, in all example discussed in this paper, including the 2013 from Bletery and Noquet, SSEs are always migrating before the coalescence. Does case "a" represent a realistic scenario ?

Regarding the numerical models presented:

It is stated in the text (l.235-236) that models without frictional heterogeneities can also generate migrating SSEs. It would be interesting to show the result of such a model. If frictional heterogeneities are not required to generate coalescing SSEs, the authors need to better explain why they choose to add additional complexity to generate those events: if they are indeed more frequent whith the addition of frictional heterogeneities, this could be discussed (with some statistics on the number of coalescing events).

Indeed, the author show one example of coalescing SSE for each model (I, II or III). It could be interesting to discuss how representative these models are, and what are the probabability of observing migrating SSEs in each model: are some frictional configurations more likely to generate coalescing SSEs ? Are they frequent or rare ? Could you quantify the probablity of observing coalescing SSEs in each scenario ? How does it compare with natural observations with natural observations ?

Minor comments:

- l.76-79: Do you really want to refer to the acceleration / deceleration of the rupture front ? I don't think this can be easily resolved, and it is not discussed in the paper of Bletery&Nocquet. I think you should rather discuss acceleration / deceleration in slip rate and/or moment rate.

-l. 240: "and event's duration" => could be clarified by "and increasing the event duration"

The images or other third party material in this Peer Review File are included in the article's Creative Commons

license, unless indicated otherwise in a credit line to the material. If material is not included in the article's Creative Commons license and your intended use is not permitted by statutory regulation or exceeds the permitted use, you will need to obtain permission directly from the copyright holder.

**Response to Reviewer 1' Comments:**

Reviewer #1 (Remarks to the Author):

Wang et al. present a study on secondary acceleration of slip fronts associated with the
coalescence of slow slip events (SSEs) in Cascadia. They perform kinematic slip
inversions of GNSS data for all events that have happened between 2012 and 2023.
They find a coalescence episode of SSEs in 2021, adding up to the one previously
identified in 2013. Confirming the findings of Bletery and Nocquet (2020) for the 2013
case, they find that this merging episode is associated with slip acceleration. They
further present 3 numerical models explaining this behavior. Additionally, they discuss
the specificity of migrating SSEs based on scaling laws.

Overall, I find the study interesting. This second observation of an SSE coalescence
deserves to be published. Nevertheless, I do have a list of comments / questions that I
hope will help improve the manuscript.

**Response:** Thanks for your encouraging comments. Below, please find our point-to-
point response to your other suggestions.

**Comment 1:** On the structure of the article:

The final part of the manuscript (on scaling laws of migrating SSEs) appears a bit
disconnected from the rest. Consider clarifying the connection to the rest of the paper.

More generally, the manuscript is a bit “messy” with 65 supplementary figures
appearing in a rather chaotic order, passed comments appearing in the middle of
sentences (see minor comments) and references appearing multiple times (e.g., ref 10
and 15).

**Response:** We added an introductory statement at the beginning of the section "The
scaling laws for SSEs" to explain the reason why we are discussing scaling laws. The
key motivation is that, unlike coalescence during the acceleration phase, coalescence

during the deceleration phase requires at least one SSE to migrate, due to the natural
 decline in slip rate and rupture area. The coalescence of SSEs in the deceleration phase
 is considered a special event of migrating SSEs. We drew a diagram to illustrate our
 viewpoint (Figure S2):

**Figure R1: Schematic diagram of the movement of the SSE rupture center along**
 **strike.** (a) The coalescence process occurs during the acceleration phase without
 migration. (b) The coalescence cannot occur during the deceleration phase without
 migration. (c) The coalescence during the deceleration phase requires at least one SSE
 migration. The yellow area represents the nucleation phase, while the orange and pink
 areas indicate the acceleration and deceleration phases, respectively.

 For this reason, we classify SSEs into migrating and non-migrating categories. We now
 explicitly state that coalescing SSEs in the deceleration phase represent a special class
 of migrating SSEs. The revised text reads (Lines 275-281):

 *" The feature of SSEs coalescence is the spatial convergence of multiple SSEs. Unlike*
 *coalescence during the acceleration phase, coalescence during the deceleration phase*
 *requires at least one SSE to migrate, due to the natural decline in slip rate and rupture*
 *area (see Fig. S1). We categorize the SSEs in the catalog into migrating SSEs and non-*
 *migrating SSEs by examining whether SSE ruptures and tremors migrate along the*
 *strike. The coalescence of SSEs in the deceleration phase is considered a special event*
 *of migrating SSEs. "*

 We added an overview of the 67 supplementary figures at the beginning of the
 supplementary document to help readers quickly understand the logic of the appendix

figures. We have now clarified this in the text with the following (Lines 40-49 in the
supplement):

*"Supplementary Figures Overview:*

1. *Figures S1. Conceptual diagram of SSEs coalescence.*

2. *Figures S2-S3. Station distribution and fault geometry of the Cascadia*
*subduction zone.*

3. *Figures S4. vbICA preprocessing of GNSS time series.*

4. *Figures S5-S6. Deep learning detection results of SSEs.*

5. *Figures S7-S16. Supplementary files for SSE 44.*

6. *Figures S17-S23. Supplementary files for numerical simulations.*

7. *Figures S24-S25. Supplementary files for scaling law.*

8. *Figures S26-S68. Supplementary files for the inversion results of all SSEs."*

The issue of duplicate references has been corrected.

**Comment 2:** On the inversion of the coalescence episode:

I see in Fig S7 that only about a third of GNSS stations have been used in the inversion.

Why?

I would strongly recommend performing the inversion on all available stations. Would
that change the results?

**Response:** Thank you for pointing this out. Due to our oversight, we forgot to update
this figure in the supplementary material, which came from a previous iteration of our
manuscript.

We have changed the figure accordingly:

**Figure R2: GNSS network used in SSE 44.** The circles mark the locations of the
 GNSS sites used for kinematic inversion. The dashed lines are the Slab2.0 iso-depth
 contours every 20 km.

Based on your suggestion, we performed inversions using all available stations and the
 stations where SSE signals were detected by the deep learning model, separately. The
 results of both station selection strategies are as follows:

**Figure R3: Inversion results for different station selection strategies.** (a) Station
 distribution detected by the deep learning model for SSE44. (b) Snapshot of the
 inversion results before and after coalescence using the machine learning catalog. (c)
 Source time function using the machine learning catalog. (d) All available stations in

the region for SSE44. (e) Snapshot of the inversion results before and after coalescence
using all available stations. (f) Source time function using all available stations.

We selected the stations in Figure R2 based on a thorough evaluation of deep learning
detection results and signal-to-noise ratios. These stations provide sufficient resolution,
as confirmed by dynamic resolution tests, and are evenly distributed to optimize
inversion accuracy. To minimize noise interference, we excluded stations near the deep
fault zone, where resolution is typically limited. As shown in Figure R3, the inversion
results remain consistent essentially across three different station selection strategies,
validating the robustness of our station selection strategies. Therefore, we choose to
retain the station selection shown in Figure R2.

**Comment 3:** On the question of “merging vs normal” SSEs:

Define what a “normal” SSE is.

I am guessing that you mean a non-migrating SSE. This brings the question of the
resolution of the kinematic inversions. Migrations are much easier to see for large SSEs
than for small ones. This could explain why migrating SSEs appear to have larger
moment (and therefore larger extent and duration) than “normal” ones. One way to
investigate whether “normal” SSEs truly do not migrate would be to look at the
associated tremor distributions. Consider strengthening the discussion on SSE
migrations based on tremor migrations.

**Response:** We thank the reviewer for highlighting this important point. You are correct:
by “normal” SSEs, we were referring to non-migrating SSEs, and we have revised the
terminology throughout the manuscript to consistently reflect this (see also revisions
related to Comment 1 and Fig. 4 in the main text).

Regarding the reviewer’s concern about observational bias: indeed, migration is more
easily resolved in larger SSEs due to their longer duration and broader spatial extent.

We agree that this could influence the classification of events. However, this

observation is also consistent with the predictions of a finite-width rupture model, as
discussed in our numerical simulation section. In such models, SSEs initially propagate
in both dip and strike directions. Once the slip front saturates the width of the velocity-
weakening (VW) zone, continued rupture can only occur along strike, giving rise to
migration. This mechanism explains why migrating SSEs tend to exhibit larger
moments, rupture areas, and durations, they are not just more visible, but physically
larger due to the along-strike growth.

To address the second part of the reviewer's comment, we fully agree that tremor
distributions provide an important, independent constraint on SSE migration. In our
analysis, we leverage both GNSS-derived slip distributions and tremor catalogs (from
PNSN) to classify SSEs. We observe that tremor migration correlates well with
geodetic migration in both space and time (see Fig. 1a and Fig. 4b), reinforcing the
validity of our classification. We now clarify this point more explicitly in the revised
text (see lines 336-365 in the discussion section), where we also emphasize the
importance of combining geodetic and seismic indicators to interpret SSE dynamics.
The updated discussion text reads:

*" Our study shows that SSEs can coalesce not only during the accelerating phase but*
*also during the deceleration phase. The 2021 event demonstrated that the coalescence*
*of the slip fronts during the deceleration phase led to a secondary acceleration of slip*
*velocity and the expansion of the rupture area, forming a secondary peak in the moment*
*release rate. This phenomenon reveals the complexity of SSE interactions, where the*
*coalescence counteracts the slip attenuation during the deceleration phase. Numerical*
*simulations suggest that the 2021 event might have been a coincidental occurrence*
*avored by regional stress heterogeneity. Coalescence during the deceleration phase*
*requires at least one SSE to migrate along the strike. Scaling law analysis indicates*
*that migrating SSEs release more energy, involve larger rupture areas, and exhibit a*
*frequency-magnitude distribution with a turning point at Mw 6.3. Coalescing SSEs can*
*be considered a specific case of migrating SSEs. While it is difficult to extract specific*
*trends in the scaling laws for these events due to their small number, the ones occurring*
*in the deceleration phase seem to be potentially longer than the other migrating events.*

*Past studies have often divided the CSZ into different segments based on cumulative*
*slip distributions, tremor distributions, and gravity anomalies^{1,2}. Our research*
*highlights that SSEs and tremors may overcome these segment boundaries due to*
*factors such as stress memory and that SSEs in different regions can coalesce. It is*
*currently clear that both coalescence during the acceleration and deceleration phases*
*lead to an increase in slip rate. However, the impact of SSE coalescence on the spatial*
*and temporal range of future large earthquakes remains unclear. We speculate that the*
*acceleration of slip rate and expansion of rupture area caused by SSE coalescence*
*could significantly alter the redistribution of fault stress, influencing the occurrence or*
*rupture process of future earthquakes. Future work will need to use more refined*
*numerical simulations and long-term observations to quantify the contribution of*
*coalescing SSEs to stress fields and earthquake cycles, assessing their role in*
*earthquake disaster forecasting. In conclusion, the coalescence of SSEs during the*
*deceleration phase highlights the role of fault heterogeneity and migration behavior in*
*the dynamics of SSEs."*

**Reference:**

(1) Li, D. & Liu, Y. Modeling slow-slip segmentation in Cascadia subduction zone
constrained by tremor locations and gravity anomalies. *Journal of Geophysical*
*Research: Solid Earth* **122**, 3138-3157 (2017).

(2) Michel, S., Gualandi, A. & Avouac, J.-P. Similar scaling laws for earthquakes and
Cascadia slow-slip events. *Nature* **574**, 522-526 (2019).

**Minor comments:**

**Comment 4:** L21-22: I think the sentence “can influence the timing and 21 spatial
extent of future seismic activity in the region. ” was added by mistake in the middle of
another sentence.

**Response:** We rewrote the abstract of the paper to better summarize the key points of
our paper (see lines 22-37).

**Comment 5:** L29: List factors.

**Response:** We rewrote the abstract of the paper to better summarize the key points of
our paper (see lines 22-37).

**Comment 6:** Fig 3a-c: What do the colors correspond to? Contrasts in D_c and
Σ_n ? Explain in the caption.

**Response:** Exactly as you thought, the colors represent the variation in heterogeneous
 properties of the VW zone along the strike for each model. We have added a color label
 to the color bar, and we have provided a detailed explanation for a-c in the caption (lines
 215-225):

 **Figure. R4 SSE numerical simulation.** (a-c) Three model configurations, each
 comprising VS ($a-b > 0$) region surrounding a central VW ($a-b < 0$) patch. **a**, in the $a-b$
 model, the VW region exhibits variations in $a-b$ along the strike. **b**, in the D_c model,
 the VW region shows variations in D_c along the strike. **c**, in the σ_n model, the VW
 region displays variations in σ_n along the strike. **(d-f)** Spatiotemporal evolution of slip
 rates following the coalescence of SSEs during the deceleration phase for the three
 models. In panel **d**, the red line represents the heterogeneous variation of $a-b$. The
 horizontal red lines in panels **b** and **c** display the edges of the barrier. **(g-h)** Temporal
 evolution of moment rate and maximum stress for the coalesced events in the three
 models. The red dashed line indicates the moment of slip front coalescence.

**Response to Reviewer 2' Comments:**

Reviewer #2 (Remarks to the Author):

Dear editor and authors,

I thank you for the opportunity of reviewing this work.

In this paper, the authors study a catalogue of slow slip events (SSE) in the Cascadia
subduction zone. The catalogue was obtained by analyzing continuous GNSS recording
lasting 14 years. They also compare these data to a catalogue of tremors. 49 SSE are
detected. For each, the slip rate evolution on the fault zone is inverted., and the stress
change evolution is estimated. The authors investigate the special case of an SSE
coalescence. They describe the spatio-temporal evolution of this SSE., and its relation
with the tremors evolution. Moreover, they propose dynamic simulation base on a rate
and state law.

The data, and data analysis, providing the catalogue, of SSE are really interesting.
Moreover, the detailed study of the coalescence event is also of interest. However, the
paper suffer from a lack of logical structure, and the findings are not satisfactorily
discussed and analyzed altogether to drive clear conclusions or interpretations.
Moreover, a number of typos, and editing problem in the figures must be corrected. To
conclude, I find that the scientific content is solid and valuable, but that the paper needs
major revision in order to convey a clear message.

**Response:** Thank you for your insightful and detailed review of our manuscript. We
greatly appreciate your recognition of its scientific value and the thorough analysis of
the coalescence event. We have carefully considered your feedback and made
substantial revisions to improve the logical structure and provide a deeper discussion
of the results, ensuring that the manuscript conveys a clearer message.

Here are some of the major points that impeded the understanding of the paper and need
to be addressed.

**Comment 1:** It is said L84 that the catalogue is available. But then in the methods and
sup mat, it is presented as if it is the core of this work to build this catalogue. So this is
unclear and should be clarified. Then, if this is the core of this work, the catalogue
should be presented at the beginning, and not at the end.

**Response:** Thank you for your valuable feedback. We provide in this work the
kinematic inversions of SSEs that were detected by our previous study (see Reference
1). We now clarified this point by modifying the first paragraph of section "*The*
*secondary peak in moment rate from slip coalescence*" (lines 89-92):

*"In this study, we examine the evolution of slip rates associated with SSEs from 2012*
*to 2022 by performing kinematic inversions of GNSS data. To identify the timing of*
*each SSE and the corresponding affected GNSS stations, we use the detection results*
*from a previous study¹ based on a machine learning model (see Methods for details)."*

**Reference:**

(1) Wang, J., *et al.* Detecting slow slip events in the Cascadia subduction zone from
GNSS time series using deep learning. *GPS Solutions* **28**, 1-16 (2024).

We have made the model openly accessible in the Code Availability section, which
facilitates readers' understanding for the detection process (lines 467-460).

*"The deep learning-based SSE detector is available at*
*https://github.com/evedor/SSE_Detector. "*

The focus of this study is to illustrate the coalescing phenomenon during the SSE
deceleration phase, not to create a comprehensive SSE inversion catalog. The inversion
catalog isn't essential for detecting coalescing events. The deep learning detection
results for SSEs (on the station signal level) can roughly reflect the evolution of SSEs
(Fig. R5). "The scaling law for SSEs" section conducts a systematic statistical analysis
of the catalog because we consider migration to be a necessary condition for the
coalescence during the deceleration phase (see lines 275-282 and Fig. R1). So, we hope
to maintain the original structure of the paper.

*Fig. R5 Evolution of deep learning detection results for the SSE coalescing event in*
 *2021. The blue dots represent stations where SSE signals were detected on that day by*
 *the deep learning model, and the red boxes indicate the area of signal aggregation for*
 *SSEs.*

**Comment 2:** It is not clearly explained how SSE are classified as “normal”, “migrating”
 or “coalescing”.

**Response:** We classified the SSEs in the catalog into migrating SSEs and non-
 migrating SSEs by examining whether the SSE rupture and tremors migrate along the
 strike. We corrected the technical term by changing 'normal' to 'non-migrating'. The
 coalescence of SSEs in the deceleration phase is considered a special event of migrating
 SSEs. We have now clarified this in the text with the following (lines 278-281):

*"We categorize the SSEs in the catalog into migrating SSEs and non-migrating SSEs*
 *by examining whether SSE ruptures and tremors migrate along the strike. The*
 *coalescence of SSEs in the deceleration phase is considered a special event of migrating*
 *SSEs."*

**Comment 3:** Overall, there seems to be some confusion between slip acceleration (an
 increase of slip rate) and front acceleration (an increase of migrating velocity of the
 fronts). Both can be linked, but when the two events have merged, there isn't anymore

fronts. So it is not clear what is the message the authors which to convey: are the slip
front accelerating before they merge? or is the slip rate increased upon coalescence?
Maybe this confusion arise from the fact that what is called a slip front is not really
described? Is it the limit of the slipping zone? or the area in which the slip rate is larger?
How are these two related?

**Response:** We thank the reviewer for this important and insightful observation. Indeed,
there are two related but distinct processes in our study:

- • *Slip acceleration*, referring to an increase in the slip rate (or moment release rate)
within the active rupture area
- • *Front acceleration*, referring to an increase in the migration velocity of the slip
front, is defined as the boundary of the actively slipping zone.

Before coalescence, each SSE displays its own front propagation along strike. During
this phase, it is possible to analyze both the migration speed of the fronts and the
corresponding changes in slip rate. However, once the two SSEs coalesce, the slipping
zones unify, and the notion of two separate slip fronts no longer applies in a meaningful
way. At this stage, we observe a secondary increase in slip rate and moment release,
accompanied by a renewed expansion of the rupture area. While this expansion can be
described as a new front propagation, it is no longer related to the original two SSE
fronts (see lines 136-145 in the main text).

To avoid reader confusion, we have added a definition of the slip front at its first
mention in the main text as follows (lines 42-45 in the main text):

*"SSEs typically progress through several stages: nucleation, along-dip saturation,*
*along-strike acceleration and deceleration of the rupture front (the boundaries of the*
*slipping region) and slip rate."*

**Comment 4:** Why do the authors chose to focus their analysis on this specific
coalescence event? Is it the only one observed besides the 2013 event? Would it be
worth to conduct the same analysis on the 2013 event, or too redundant with previous
studies. Are these the only 2 coalescing events in the catalogue?

**Response:** The 2021 event is noteworthy because it occurred during a deceleration
phase, in contrast to the acceleration phase of the 2013 event. In contrast to coalescence
during the acceleration phase, the slip rate increase observed during the deceleration
phase can be more confidently attributed to the process, since acceleration at this stage
is not part of the natural SSE evolution. Although the methods are similar, we believe
the coalescence during the deceleration phase has unique characteristics, contrasting
with the 2013 event.

In 2019, there was also a coalescing event during the deceleration phase (Fig. S53 and
Fig. R6), occurring at a location like the 2021 event. Although the 2019 SSE inversion
results are influenced by the Laplace smoothing parameter, the spatial and temporal
distribution of the tremors also indicate that the 2019 SSE is a coalescing event in the
deceleration phase (Fig. R6c). Both events have similar locations, potentially
suggesting that similar physical dynamics were at play (Fig. R6a). Additionally, each
event displays a bimodal moment rate before and after coalescence, indicating they
occurred during the deceleration phase (Fig. R6b). Since the bimodal feature is more
pronounced in the 2021 event, the 2021 event is chosen as the representative to illustrate
my point.

**Fig. R6. Comparison of SSEs in 2019 and 2021.** (a) Accumulative slip distribution,
 with the red solid line representing the cumulative slip of the 2019 SSE, and the brown
 solid line representing the cumulative slip of the 2021 SSE. The red dashed line
 indicates the latitude of SSE coalescence in 2019, and the brown solid line indicates
 the latitude of SSE coalescence in 2021. (b) Moment rate curves for the two events, with
 the gray dashed line indicating the coalescing time. (c) Snapshots before and after SSE
 coalescence in 2019, with black scatter points representing the distribution of tremors.

We added a description of the 2019 event at the beginning of the "Simulation of SSE
 coalescence and influencing factors" section (see line), leading into the discussion of
 the physical influencing factors (see lines 192-198):

" The deep learning-based SSEs catalog and GNSS data inversion have revealed SSEs
 coalescence and the resulting slip acceleration. Aside from the coalescing event in 2021,
 a similar coalescing event during the deceleration period also occurred at the same
 location in 2019, with the moment rate showing also a bimodal characteristic (see Fig.
 S55). This implies that the physical parameters of the region may influence or control
 the repeated occurrence of such coalescing events. Next, we will investigate the factors
 that led to the coalescence during the deceleration period. "

Finally, we added the location of the 2019 event in the figure of the scaling law section

(see Fig. 4).

**Fig. R7 Spatio-temporal distribution of SSES and the scaling laws of SSES.**

**Comment 5:** It is absolutely not clear why the numerical model were conducted, why

heterogeneity was introduced, and most of all, how they compare to the data and/or

help to understand the observations. They look like they are interesting by themselves
but are not discussed.

**Response:** We have now included the purpose and rationale for introducing numerical
simulations and heterogeneity, see the first paragraph of the section 'Simulation of SSE
Coalescence' as follows (lines 198-207):

*"Numerous observational studies¹⁻³ and simulation studies^{4,5} indicate that*
*heterogeneous features, such as friction parameters and stress states, play a critical*
*role in controlling SSE behavior. Furthermore, some studies suggest that the velocity-*
*weakening (VW) zone of the CSZ exhibits heterogeneous properties. For instance,*
*Michel et al.⁶ proposed an along-strike segmentation of the CSZ into 13 distinct*
*segments based on the extent of their detected SSEs from GNSS data, while Li and Liu⁷*
*have segmented the CSZ into 5 segments based on gravity anomalies, which they used*
*to infer a distribution of effective normal stress parameters for their numerical*
*simulation of SSEs. We here developed rate-and-state friction models to examine SSE*
*coalescence and its controlling factors (Fig. 3), using quasi-dynamic (QDYN)*
*simulations⁸ with a modified rate-and-state friction (RSF) law."*

**Reference:**

- (1) Barnes, P.M., et al. Slow slip source characterized by lithological and geometric
heterogeneity. *Science Advances* **6**, eaay3314 (2020).
- (2) Saffer, D.M. & Wallace, L.M. The frictional, hydrologic, metamorphic and thermal
habitat of shallow slow earthquakes. *Nature Geoscience* **8**, 594-600 (2015).
- (3) Wang, K. & Bilek, S.L. Invited review paper: Fault creep caused by subduction of
rough seafloor relief. *Tectonophysics* **610**, 1-24 (2014).
- (4) Skarbek, R.M., Saffer, D.M. & Savage, H.M. Not all heterogeneity is equal: Length
scale of frictional property variation as a control on subduction megathrust
sliding behavior. *Geophysical Research Letters* **52**, e2025GL115738 (2025).
- (5) Skarbek, R.M., Rempel, A.W. & Schmidt, D.A. Geologic heterogeneity can
produce aseismic slip transients. *Geophysical Research Letters* **39**(2012).
- (6) Michel, S., Gualandi, A. & Avouac, J.-P. Similar scaling laws for earthquakes and
Cascadia slow-slip events. *Nature* **574**, 522-526 (2019).

(7) Li, D. & Liu, Y. Modeling slow-slip segmentation in Cascadia subduction zone
 constrained by tremor locations and gravity anomalies. *Journal of Geophysical*
 *Research: Solid Earth* **122**, 3138-3157 (2017).

(8) Luo, Y., Ampuero, J. P., Galvez, P., Van den Ende, M. & Idini, B. QDYN: a Quasi-
 DYNAMIC earthquake simulator (v1. 1). (2017).

We then added a brief subsection at the end of the section "Simulation of SSE
 Coalescence" to discuss how numerical simulations contribute to understanding the
 observed coalescence event in 2021. Based on the bimodal interval of the moment rate
 curve and the migration rate of the slip front, we think that Models I and III are the most
 likely to explain the 2021 coalescence event observations. Furthermore, by comparing
 the forearc structure with the location of the 2021 event, it is likely that the fault extends
 to a major thrust fault in this region, which causes fluid escape, thereby reducing the
 fluid pressure at the plate boundary (Fig. R8).

**Fig. R8 Forearc structure of the Cascadia subduction zone (adapted from Wells et**
 **al. 5).** A: Faults define boundaries of tremor termination and initiation. B: The blue box

indicates the location of the 2021 event. The northern part lies beneath the central
Oregon segment with low tremor density, associated with Siletzia, the accreted oceanic
basalt terrane composing the Oregon forearc. In contrast, the southern part corresponds
to the Klamath Mountains, where the deep structure involves lower velocity, quartz-
rich sediments of the Franciscan accretionary complex. The fault may extend to a major
thrust fault, providing a route for fluid escape (blue arrows), thus reducing fluid
pressure at the plate boundary and resulting in heterogeneous normal stress on both
sides of the event location.

We proposed that the 2021 coalescence event was most likely an occasional event
favored by stress heterogeneity, but the frictional heterogeneity on both sides of the
fault cannot be disregarded (lines 251-276).

*" We now compare the results of the three models with the actual observations of the*
*2021 coalescing SSE. In Model I, the moment rate reaches its peak almost immediately*
*after SSE fronts coalesce and decays rapidly, consistent with the rapid release of*
*instantaneous slip rate during the acceleration phase coalescence observed by Bletery*
*and Nocquet¹. In contrast, Models II and III display a notable time delay of the moment*
*rate peak after the coalescence. Model II reaches its peak approximately 18 days after*
*the coalescence, while Model III attains a secondary peak around 6 days, aligning with*
*the observed moment rate peak on May 4 following the SSE coalescence on April 28,*
*2021. Note that the actual values of D_c and σ_n might control the delay between the*
*start of the coalescence and the moment rate peak. Regarding rupture propagation*
*velocity, Model I concurs with observations, exhibiting distinct migration speeds for*
*the two SSEs (Fig. S23). As mentioned, Models II and III show secondary variations in*
*migration velocity due to barrier effects (differences in D_c or σ_n). It is consistent with*
*the observed slowdown in the propagation velocity of tremors along the strike = as*
*tremors approach the coalescing zone (Fig. S23). Note that the 2021 SSE occurred*
*around 44°N, where the northern part lies beneath the central Oregon segment with*

*low tremor density, associated with Siletzia, the accreted oceanic basalt terrane*
*composing the Oregon forearc^{2,3}. The southern part corresponds to the Klamath*
*Mountains, where the deep structure involves lower velocity, quartz-rich sediments of*
*the Franciscan accretionary complex⁴. These geological differences result in*
*variations in frictional properties and critical slip distance between the two regions.*
*Wells et al.⁵ suggested that the fault in this region extends to an overpressured*
*megathrust, providing fracture channels for fluid escape into the upper plate, thus*
*reducing fluid pressure and resulting in heterogeneous normal stress. We propose that*
*the 2021 coalescence event could have occurred as a result of a combination of those*
*factors, but further analysis is needed."*

**Reference:**

- (1) Bletery, Q. & Nocquet, J.-M. Slip bursts during coalescence of slow slip events in
Cascadia. *Nature communications* **11**, 2159 (2020).
- (2) Brudzinski, M.R. & Allen, R.M. Segmentation in episodic tremor and slip all along
Cascadia. *Geology* **35**, 907-910 (2007).
- (3) Audet, P. & Bürgmann, R. Possible control of subduction zone slow-earthquake
periodicity by silica enrichment. *Nature* **510**, 389-392 (2014).
- (4) Calvert, A.J., Preston, L.A. & Farahbod, A.M. Sedimentary underplating at the
Cascadia mantle-wedge corner revealed by seismic imaging. *Nature Geoscience*
**4**, 545-548 (2011).
- (5) Wells, R.E., Blakely, R.J., Wech, A.G., McCrory, P.A. & Michael, A. Cascadia
subduction tremor muted by crustal faults. *Geology* **45**, 515-518 (2017).

**Comment 6:** A global, analytical discussion of all the findings is needed, as well as a
clear conclusion.

**Response:** Following your suggestion, we have added a discussion section at the end
of the main text to summarize the study's findings and connect the results from all
sections (lines 336-365).

**Comment 7:** A lot of figure in the supplementary material cannot be read entirely
because the lack the resolution to do so even when zooming on a screen computer.

**Response:** We have replaced some figures with PDF format. However, we think that
the main reason is that during system submission, we compressed the supplementary
file to meet the size requirements of the submission system. we compressed the
supplementary file to <50MB, which may have caused the figures in the supplementary
file to become unclear. We will try to submit the original files through the submission
system.

**Comment 8:** The detailed methods can stay in the Methods section, but a clear simple
summary of the work conducted is needed in the main text, at the beginning. As is, the
reader cannot understand how the slip rate and stress changes curves come from.

**Response:** We employed finite fault inversion to calculate the slip rate. Stress changes
were resolved onto receiver faults with specified slip directions and geometries¹.
Additionally, we calculated the stress rate from the slip rate using a 3D elastic
dislocation model², focusing on the shear component. Specifically, we utilized the
Green's function generated by the EDCMP program³ and the slip rate inverted from
each fault in this study.

To enhance reader comprehension, we have added a brief explanation of the methods
at the beginning of the first and second sections.

**References:**

(1) Geoffrey C. P. King, Ross S. Stein, Jian Lin; Static stress changes and the
triggering of earthquakes. Bulletin of the Seismological Society of America. 84 (3):
935–953 (1994).

(2) Yoshimitsu Okada; Internal deformation due to shear and tensile faults in a half-
space. Bulletin of the Seismological Society of America. 82 (2): 1018–1040 (1992).

(3) Wang, R., Martín, F.L. & Roth, F. Computation of deformation induced by
earthquakes in a multi-layered elastic crust—FORTRAN programs
EDGRN/EDCMP. Computers & Geosciences 29, 195-207 (2003).

Here are some minor typo or problems that should also be addressed or corrected..

**Comment 9:** there is a piece of sentence that should not be here

**Response:** We rewrote the abstract of the paper to better summarize the key points of
our paper (see lines 22-37).

**Comment 10:** What is called a proto SSE?

The translation was incorrect, so we removed 'proto'.

**Comment 11:** L29 can you state the factors, or the most important ones?

**Response:** We rewrote the abstract of the paper to better summarize the key points of
our paper (see lines 22-37).

**Comment 12:** L34: “which can modify stress”: remove can, it does modify stress.

**Response:** Done.

**Comment 13:** figS1 B: The sketch is not really clear to me, because if there is merging,
there should be 2 SSE depicted. S1b: accelerate faster is not clear. do you mean higher
acceleration, do you mean (accelerate more=higher acceleration) or (time derivative of
the acceleration grows) .

**Response:** What we mean is "higher acceleration". We have redrawn a new schematic,
with each subplot containing two SSEs (see Fig. R1). The diagram describes the
interaction of SSEs under three different scenarios, focusing on the slip burst
phenomenon during the acceleration phase and the secondary acceleration phenomenon
during the deceleration phase. Additionally, Figures b and c of Fig. R1 emphasize that
the necessary condition for the coalescence during the deceleration phase is SSE
migration, which better illustrates why we classified SSEs as migrating and non-
migrating in the Scaling Law section.

**Comment 14:** word or end of sentence missing.

**Response:** Done.

**Comment 15:** L84: is the catalogue available, or have you built it as explained in
sup.mat.

**Response:** The raw catalog provided in our previous paper only included time points
and the quantity of stations. This paper offers a more detailed station distribution, and
we have repeatedly validated it through inversion, as explained in Comment 1.

**Comment 16:** test S2 L56 60: is this output probability of SSE or occurrence and
duration of SSE.?

**Response:** The model's output represents the probability of SSE at each time point. For
example, if the probability exceeds 0.98 for each time point between January 1st and
February 1st for more than four stations, then we can conclude that an SSE occurred
between January 1st and February 1st. We have now clarified this in the text with the
following (lines 100-101 in the Supplement):

*"The model's output represents the probability, ranging from 0 to 1, that each station*
*is classified as experiencing an SSE at each time point."*

**Comment 17:** fig S3: color scale cannot be read, improve resolution or mak it bigger.

Done. As explained in Comment 7.

**Comment 18:** fig S4: SSE 22 appears two time SSE 23 is missing.

Done. Thank you for your very careful reading.

**Comment 19:** Fig 1: What is the difference between a near field tremor and a total
tremor. Why are they not discussed in text?Why do tremors remains active in the north
while no slow-slip is detected?

**Response:** The number of tremors is derived from the distance (within 50km) between
the spatiotemporal distribution of the tremors and the inversion slip. The tremors in the
southern part of Cascadia are constantly occurring, as seen in Fig. 4(a), where tremors
are continuously recorded in the region between 40 – 44°N. However, It does not
always capture the geodetic signals of SSEs. A similar phenomenon can be observed in
the deep regions of New Zealand¹, where deep tremors remain active even outside long-
term SSE periods. This is a common phenomenon in these specific areas. Since it is not
the focus of the study, we did not discuss in the main text.

**References:**

(1) Aden-Antoniów, F., et al. "Low-frequency earthquakes downdip of deep slow slip
beneath the North Island of New Zealand." *Journal of Geophysical Research:*
*Solid Earth* 129.5 (2024): e2023JB027971.

**Comment 20:** Fig 2: text and label a little bit too small. There are editing problems in
fig 2a (grey boxes around label

**Response:** We have enlarged the scale values, labels, and legends, as shown in the
figure below (lines 181-190):

**Fig. R9 Stress and tremor count variation during coalescence of SSE in the**
 **deceleration phase. a,** Stress changes from April 27th to May 2nd. The blue line
 represents the number of tremors, the red line represents the slip rate, and the yellow
 line represents the stress release rate along the strike. The red arrows indicate the slow
 slip fronts propagation direction, and the purple arrows represent the increase or
 decrease in tremor counts in the coalescing area. b, Schematic diagrams of tremor and
 slip in the early stage of two SSEs coalescence. c, Schematic diagrams of tremor and
 slip in the intermediate stage of two SSEs coalescence.

**Comment 21:** L138-140: I find that the description does not corresponds to the figure.
 The tremor activity looks almost constant during the whole second part, and I do not
 think that the peak during the lower slip rate is minor, but rather larger.

**Response:** Our description may have been unclear, so we have rewritten this section.
 Tremors during this period were not constant. The unit of tremors (as shown in the
 legend in the upper right corner of the figure R4) is in 10^3 . Thus, a change of 0.3 in
 Figure 2b represents an increase of 300 detected tremors compared to the previous day.

The peak of tremors during the lower slip rate is larger. This is what we intended to
convey. We have now rewritten this in the text with the following (lines 165-180):

*" During the onset of slip coalescence, two slip fronts converge in the coalescing zone,*
*leading to a peak in tremor activity. This peak is driven by elevated slip rates and the*
*combined shear stress transients from both slip fronts, which are sufficient to trigger*
*tremors due to their high sensitivity to minor stress changes (e.g., 1–2 kPa/day,*
*comparable to tidal or teleseismic stresses)^{1,2}. Tremors are predominantly located*
*within the coalescing zone, positioned between the two regions of maximum stress*
*release (Fig. 2b, c). As the coalescing process progresses, slip rate in the coalescing*
*zone gradually increase, while stress release rates continue to rise. There is a decrease*
*of tremor count (36% of decrease relative to the peak amount of tremor) between the*
*29th and 30th of April (Fig. 2a), but then it seems to roughly stabilize the following*
*days. The decline occurs because tremors reach a seismic saturation point after*
*releasing approximately 25 kPa of stress. Beyond this threshold, their sensitivity to*
*additional stress changes decreases, even as aseismic slip continues³. The increase in*
*tremor count occurs at latitude 43.5°N. At the same location, the tremor count*
*continues to decrease in the following days while the moderate tremor peak has*
*migrated to the north at latitude 44°N."*

**Reference:**

(1) Rubinstein, J.L., et al. Seismic wave triggering of nonvolcanic tremor, episodic
tremor and slip, and earthquakes on Vancouver Island. *Journal of Geophysical*
*Research: Solid Earth* **114**(2009).

(2) Houston, H. Low friction and fault weakening revealed by rising sensitivity of
tremor to tidal stress. *Nature Geoscience* **8**, 409-415 (2015).

(3) Schwartz, S.Y. & Rokosky, J.M. Slow slip events and seismic tremor at circum-
Pacific subduction zones. *Reviews of Geophysics* **45**(2007).

**Comment 22:** L141. I do not understand this interpretation. From the data it looks like
the tremor are located inside the high slip rate area, so why do you state that they occur
in a locked area?

**Response:** Our expression caused some misunderstanding. What we meant was that
the tremors are located within the high slip-rate area in the subduction zone. We have
rewritten this section for better clarity in the explanation. As explained in Comment 21.

**Comment 23:** L147-151: what is the analysis that gives you the shear stress rate, where
is it presented?

**Response:** The stress changes can be resolved on to receiver faults with specified slip
direction and geometry. Here, we further calculated the stress rate by slip rate with a
3D elastic dislocation model and used the shear part. Specifically, we used the Green's
function generated by the EDCMP program and the slip rate inverted from each fault
in this study. We added this information in the main text (see lines 150-160):

*"We specifically calculate shear stresses aligned with the slip direction by EDCMP
program."*

**Comment 24:** L153-166: The description is unclear

**Response:** We have rewritten this section (see Comment 21).

**Comment 25:** L164: I did not understand what is the physical value corresponding to
25kPa?

**Response:** Tremors are sensitive to stress at the kilopascal level. Small stresses, such
as those from distant earthquakes or tidal stresses, can trigger tremors. During SSE
period, the stress is approximately 1 to 2 kPa per day, and the stress release rate we
calculated is of a similar magnitude to the tidal stress and distant earthquake stress that
can generate tremors. However, the cumulative number of tremors and the cumulative
stress release do not follow a linear relationship. After reaching a threshold, the number
of daily tremors no longer increases with increasing stress. I have rewritten this section
for better clarity in the explanation (see Comment 21).

**Comment 26:** L177-183: see main remarks number 5

**Response:** Done.

**Comment 27:** L215-249. At this point, it is unclear if you are describing simulation
results or the actual catalogue. See main remarks number 1. The results are presented
but not interpreted

**Response:** We used the actual catalogue and provided a more detailed description at
the beginning of this section (Lines 275-283). Additionally, following your suggestion,
we added a discussion section (see comment6 and lines 336-365).

**Comment 28:** L220: define T c g M_0

**Response:** We have added explanations for T_c , g , and M_0 . We have now clarified this
in the text with the following (lines 286-288):

*"where T is the duration, M_0 is the event moment defined as the integral of slip over*
*the rupture area multiplied by the shear modulus, and c and g represent the slope*
*and intercept, respectively, in the relationship between seismic moment and duration."*

**Comment 29:** Figure 4.4a: improve resolution, what are grey dots? why are certain part
in color,...? 4f include legend in the figure.

**Response:** We switched to PDF format to improve resolution. The gray dots represent
the corresponding daily tremor locations from the PNSN catalog as described in the
figure caption. Certain parts in color indicate the timing and rupture extent of the SSEs.
We have added a legend for Figure 4f.

I hope that this review comes out helpful for the authors and I thank again the editor
and the authors for the opportunity of reviewing this work.

**Response:** Thank you very much for your careful reading and insightful scientific
suggestions. The revised version of the paper, based on your suggestions, is indeed
more rigorous and has a deeper scientific significance than the previous version.

**Response to Reviewer 3' Comments:**

Reviewer #3 (Remarks to the Author):

This paper focusses on the analysis of coalescence SSEs in the Cascadia subduction
zone. It performs a detailed analysis of one event in 2021, extracted from previously
published SSE catalog, and also numerical simulations to estimate the impact of various
frictional properties on SSEs characteristics, and a discussion on SSE scaling laws.

While there are several interesting results and discussion in the paper, the link between
the different aspects discussed is not clear, and as a result I find the messages of the
paper difficult to follow.

**Response:** We appreciate your insightful suggestion and the opportunity to further
clarify the relationships between the different sections and the conclusion. Below please
find our point-by-point response.

I detail below my main comments regarding the 3 main points discussed in that paper:

The first one, related to the detailed analysis of the 2021 coalesced ETS event, is well
detailed, and brings interesting observational constraints on the relationships between
tremors and slip, and on the moment rate evolution when two SSEs merge. This study
is rather clear and well detailed.

**Response:** Thank you very much for your careful reading.

The second point related to the rate and state modeling of SSE cycles and factors
controlling the coalescence: the results presented are interesting, and different
behaviors are observed depending on the assumed frictional properties.

**Comment 1:** However, the paper only provide a rather superficial description of the
results of the different models, but does not provide any clear interpretation, nor relate
the models to the observed coalesced SSE in Cascadia. I would have appreciated a
discussion on which model (I, II, III) better explained the observed ETS sequences in
Cascadia. Which moment rate functions are more similar to the observations? The
models also seem to exhibit very different amount of slip per event and various rupture
velocity (from fig. S18), these points are not at all discussed in the paper. It is thus
difficult to assess how the numerical simulation help to understand the coalescence
process.

**Response:** Thank you very much for your suggestions. We greatly value your feedback.
As you correctly pointed out, the original version emphasized the mechanics of each
model individually but did not adequately discuss how they compare to the 2021
coalescence event or how they help interpret the observed sequence. To address this,
we added a new paragraph at the end of the ‘Simulation of SSE Coalescence’ section,
where we compare the moment rate curves, rupture propagation speeds, and spatial
patterns from Models I–III to the 2021 event (see lines 235-272 in the main text). In
addition, structural observations of the southern Cascadia forearc indicate that the fault
in this region intersects a major thrust system, facilitating fluid escape into the upper
plate and reducing pore pressure along the subduction interface. This spatial correlation
supports the idea that variations in effective normal stress, captured in Model III, may
play a critical role in controlling SSE coalescence at this location (see revised Fig. R8).

We propose that the 2021 coalescence event could have occurred as a result of a
combination of those factors, but further analysis is needed (lines 247-272). You have
raised a concern, also pointed out by Reviewer #2, see our response above to his/her
Comment 5.

**Comment 2:** The third point discussed relates to the scaling laws for SSEs: once again,
I find it difficult to relate this point with the previous ones. There are only two examples
of coalesced SSEs in the catalog (they are among the largest ETS observed), and the
discussion here is more focused on the transition between migrating / non-migrating
SSEs, and it is not clear to me what it brings to the understanding of the coalesced SSEs.
The scaling figure (Fig.4) also brings several questions, not discussed in the paper, and
listed below:

**Response:** We have added an introductory statement at the beginning of the section
"The scaling laws for SSEs" to explain the reason why we are discussing scaling laws.
The main reason is that, unlike the coalescing of SSEs in the acceleration phase, in the
deceleration phase, as the slip rate and rupture area decrease, the necessary condition
for coalescing SSEs is migration. Therefore, we have classified SSEs into migrating
and non-migrating categories. The coalescence of SSEs in the deceleration phase is
considered a special event of migrating SSEs. We have now clarified this in the text
with the following (Lines 275-281):

*" The feature of SSEs coalescence is the spatial convergence of multiple SSEs. Unlike*
*coalescence during the acceleration phase, coalescence during the deceleration phase*
*requires at least one SSE to migrate, due to the natural decline in slip rate and rupture*
*area (see Fig. S1). We categorize the SSEs in the catalog into migrating SSEs and non-*
*migrating SSEs by examining whether SSE ruptures and tremors migrate along the*
*strike. The coalescence of SSEs in the deceleration phase is considered a special event*
*of migrating SSEs. "*

**Comment 3:** Some of the events in the current study are also present in Michel et al.'s
catalog: did the author made a systematic comparison of moment and duration from
both catalogs for similar events ? Are the results consistent ?- what are the there more
small events in Michel et al.'s catalog's with respect to the current one ?

**Response:** Thank you for the valuable suggestion. We performed a systematic
comparison of the two catalogs, focusing on overlapping events, and the moment–

duration characteristics of common SSEs. The results are now summarized in the Figure
R10. We believe this comparison demonstrates the consistency of the two catalogs.

Key findings include:

- • **Overlap:** It's difficult to compare catalogs because they don't have the same
detection threshold. Events in one catalog can be divided into multiple ones in
another catalog. We can only compare events that are roughly similar in both
catalogs (Fig. R11a). More than 80% of the events in both catalogs were detected
jointly. The SSEs that were not shared mainly occur in the southern region below
44°N . In this region, SSE events have smaller moment and shorter duration,
making the signal more susceptible to noise, which leads to differences between
the two detection methods.
- • **Moment and duration comparison:** For the common events, we calculated the
kernel density distributions of moment and duration for both catalogs (see Fig.
R11b). Michel et al.'s catalog reports an average log moment of 18.1 ± 0.59 and log
duration of 6.49 ± 0.18 (seconds), while our results yield 18.48 ± 0.47 and $6.34 \pm$
0.17 , respectively. These differences fall within the expected variability,
considering the methodological differences in inversion and catalog construction.

Fig. R10 The consistency of the two catalogs. **a**, The spatiotemporal distribution of common SSEs, with slip rates normalized. Gray dots represent the spatiotemporal distribution of tremors from the PNSN catalog. **b**, Probability density distribution in the moment and timing of common SSEs between the two catalogs.

**Comment 4:** what is the origin of the different between the scaling from Michel et al.'s
 catalog and the current scaling (in particular, why did they not observe an inflection
 point?).

**Response:** Possible influencing factors to consider:

1. Parameter selection of vbICA decomposition is associated with the number and
 selection of retained components.

2. SSE detection method: Based on the ICAIM approach (Gualandi et al., 2016),
 Michel et al. reconstructed the slip deficit history along the subduction interface.

They applied temporal smoothing to the slip deficit time series to derive the
 evolution of the slip deficit rate on the fault. SSEs were then identified by applying

a slip rate threshold to this time series. The retrieved SSEs catalog was further
examined (removal of potential artifacts and merging of certain events) and led to
a catalog with 40 SSEs during the 2007-2017 period. In contrast, we used a
machine learning approach to first roughly detect and estimate the timing of SSEs,
before inverting for their kinematics with temporal smoothing. Therefore, the SSEs
detections by Michel et al. are influenced by the temporal smoothing and the
detection slip rate threshold applied, while our method is primarily affected by the
performance of the deep learning model.

3. Selection of inversion parameters for SSEs: The smoothing parameter selections
differ slightly between the two methods. However, both methods use the L-curve
approach to select the smoothing factor. While this has led to some differences in
the results, I believe both approaches are reasonable.

We think that Michel et al.'s catalog also exhibits a similar inflection point, ranging
between 6.3 and 6.4. However, since Michel et al.'s paper primarily focuses on similar
scaling laws for earthquakes and Cascadia slow-slip events, this point was not discussed.

The inflection point in Michel et al.'s catalog is shown in the figure below:

**Fig. R11** G-R law of Michel et al.'s catalog. The red dot represents the possible
inflection point in Michel et al.'s catalog.

**Comment 5:** I also did not get the criteria to consider a SSE as “normal” or “migratory”.

**Response:** We have changed "normal" to "non-migrating" to indicate that this is an
SSE that does not migrate along the strike. We categorize the SSEs in the catalog into
migrating SSEs and non-migrating SSEs by examining whether SSE ruptures and
tremors migrate along the strike. We have now clarified this in the text with the
following (lines 278-281):

*"We categorize the SSEs in the catalog into migrating SSEs and non-migrating SSEs*
*by examining whether SSE ruptures and tremors migrate along the strike. The*
*coalescence of SSEs in the deceleration phase is considered a special event of migrating*
*SSEs."*

**Comment 6:** While there are several interesting results and discussion in the paper the
link between the different aspects is not clear, points 2 and 3 lack some details, it is not
clear what they bring to the understanding on the coalesced SSEs. As a result, I find the
message(s) of the paper difficult to follow.

At its current stage, the paper gives the impression of separate studies being combined
into a single manuscript without a clear unifying message. If points 2 and 3 are intended
to contribute to a broader discussion on coalesced SSEs, this should be made more
explicit. Otherwise, the authors might consider dividing the manuscript to present a
clearer and more focused message for each component.

In my opinion, the structure and central message of the paper needs to be clarified
before it can be considered for publication.

**Response:** Regarding points 2 and 3 lacking some details, we have added
corresponding discussions and connecting paragraphs in the main text. To be more
specific, Point 2 is used to discuss the factors influencing coalescence. We have added
a description at the beginning of the section "Simulation of SSE Coalescence and
Influencing Factors" to explain why numerical simulations are conducted (lines 192-
201). At the end of this section, we included a discussion on which model is more
suitable for explaining the 2021 coalescence event. Point 3 treats the coalescing event
during the deceleration phase as a special case of migrating SSEs, with a discussion on

the distribution of the scaling law. In the section "The Scaling Law for SSEs," we have
added a connection between the scaling law and the previous content, as referenced in
comment 2. Following your suggestion, we have also added a discussion section at the
end of the main text to summarize the study's findings and connect the results from all
sections. We have now clarified this in the text with the following (lines 336-365):

*"Our study shows that SSEs can coalesce not only during the accelerating phase but*
*also during the deceleration phase. The 2021 event demonstrated that the coalescence*
*of the slip fronts during the deceleration phase led to a secondary acceleration of slip*
*velocity and the expansion of the rupture area, forming a secondary peak in the moment*
*release rate. This phenomenon reveals the complexity of SSE interactions, where the*
*coalescence counteracts the slip attenuation during the deceleration phase. Numerical*
*simulations suggest that the 2021 event might have been a coincidental occurrence*
*favored by regional stress heterogeneity. Coalescence during the deceleration phase*
*requires at least one SSE to migrate along the strike. Scaling law analysis indicates*
*that migrating SSEs release more energy, involve larger rupture areas, and exhibit a*
*frequency-magnitude distribution with a turning point at Mw 6.3. Coalescing SSEs can*
*be considered a specific case of migrating SSEs. While it is difficult to extract specific*
*trends in the scaling laws for these events due to their small number, the ones occurring*
*in the deceleration phase seem to be potentially longer than the other migrating events.*

*Past studies have often divided the CSZ into different segments based on cumulative*
*slip distributions, tremor distributions, and gravity anomalies^{1,2}. Our research*
*highlights that SSEs and tremors may overcome these segment boundaries due to*
*factors such as stress memory and that SSEs in different regions can coalesce. It is*
*currently clear that both coalescence during the acceleration and deceleration phases*
*lead to an increase in slip rate. However, the impact of SSE coalescence on the spatial*
*and temporal range of future large earthquakes remains unclear. We speculate that the*
*acceleration of slip rate and expansion of rupture area caused by SSE coalescence*
*could significantly alter the redistribution of fault stress, influencing the occurrence or*
*rupture process of future earthquakes. Future work will need to use more refined*
*numerical simulations and long-term observations to quantify the contribution of*
*coalescing SSEs to stress fields and earthquake cycles, assessing their role in*
*earthquake disaster forecasting. In conclusion, the coalescence of SSEs during the*
*deceleration phase highlights the role of fault heterogeneity and migration behavior in*
*the dynamics of SSEs."*

**Reference:**

(1) Michel, S., Gualandi, A. & Avouac, J.-P. Interseismic coupling and slow slip events
on the Cascadia megathrust. *Pure and Applied Geophysics* 176, 3867-3891
(2019).

(2) Li, D. & Liu, Y. Modeling slow-slip segmentation in Cascadia subduction zone
constrained by tremor locations and gravity anomalies. *Journal of Geophysical*
*Research: Solid Earth* 122, 3138-3157 (2017).

**Minor comments:**

**Comment 5:** 1.21 “evoluti” => evolution

**Response:** Done.

**Comment 6:** 1.22 “region. on of slip” => check sentence

**Response:** Done.

**Comment 7:** 1.51 “elevating seismic”: missing word ? please clarify

**Response:** In this version, we think that this sentence may not add much value, so we
have removed it.

**Comment 8:** 1.71-72 “and is different from the 2013 event”=> the 2023 event (ref. 15
Bletery & Nocquet) has not been discussed yet so the comparison is not very relevant.
Maybe discuss or provide a reference for this event earlier.

**Response:** We have now clarified this in the text with the following (lines 76-78):

*"Unlike the 2013 SSE coalescing event¹, which occurred during the acceleration phase,*
*the 2021 event occurred when the two initially separated SSEs were in the deceleration*
*phase."*

**Reference:**

(1) Bletery, Q. & Nocquet, J.M. Slip bursts during coalescence of slow slip events in
Cascadia. *Nat Communications* 11, 2159 (2020).

**Comment 9:** Figure 2a: it is unclear what are the units for each plots. Are they
normalized ?

**Response:** The unit is in the top-right corner of the legend.

In supplements:

**Comment 10:** l. 37: “we employ three times of vbICA” => check grammar.

**Response:** “we employ three times of vbICA” => “we use vbICA three times”

**Response to Reviewer 1' Comments:**

Reviewer #1 (Remarks to the Author):

I am satisfied with the revisions made by the authors.

I recommend publication pending minor revisions (listed below) that I believe the
authors can do without going through an additional round of review.

Minor comments:

**Comment 1:** The quality of writing is good enough for someone familiar with the
topic to follow but might be an obstacle for a broader audience. Re-writing with the
help of a native english speaker would help broadening the impact of the paper.

**Response:** Thank you for your suggestion. We have now polished the language by
professional English editing service.

**Comment 2:** The 2019 event could be discussed a bit more in the main text.

**Response:** We add more discussion on the 2019 events in the discussion section. The
focus is on comparing the 2013 northern coalescing event with the coalescing events
of 2019 and 2021 in the southern region (Lines 319-324):

*"However, the coalescing characteristics of SSEs vary across regions. The SSE*
*coalescing events in the northern CSZ (2013 events) occur during the acceleration*
*phase, leading to slip bursts phenomena. In contrast, the SSE coalescing events in the*
*southern CSZ (2019 and 2021 events) occur during the decelerating phase, showing*
*clear migration behavior, that results in a secondary acceleration of the moment rate."*

**Comment 3:** From Fig 4c, I get the impression that migrating and non-migrating
SSEs might follow different scaling laws (with $c = 1$ and $c = 3$, respectively). I would
recommend to fit c values for both populations individually and to discuss the results
(if correct this would explain the question of the scaling laws by the saturation of the
width of the transition zone).

**Response:** We performed separate fits for the two populations, and the results showed
that the c -values for the two populations are 1.5 and 3.6, respectively, which can be
explained using the bounded rupture model. These results are also consistent with the
conclusions of Gomberg, et al¹. The revised text reads (Lines 260-269):

*" We then fit the c values for migrating SSEs and non-migrating SSEs individually. The*
*results show that the c value for migrating SSEs is 1.5, while that for non-migrating*
*SSEs is 3.6. Gomberg, et al.¹ found that when the rupture reaches slip zone boundary*
*and the 2-D rupture growth transitions 1-D, the c value changes from 1 to 3. Most non-*
*migrating SSEs do not reach the slip-zone boundary, whereas a small portion reaches*
*the boundary but without migration of the source center. Therefore, their c values are*
*greater than 1 but less than 3, and closer to 1. On the other hand, migrating SSEs*
*generally reach the slip boundary, and their c values are greater than 3, reflecting the*
*physical differences that control 1-D propagation. "*

**Reference:**

(1) Gomberg, J., Wech, A., Creager, K., Obara, K. & Agnew, D. Reconsidering
earthquake scaling. *Geophysical Research Letters* 43, 6243-6251 (2016).

I hope this helps.

**Response:** Thank you for your suggestions.

**Response to Reviewer 2' Comments:**

Reviewer #2 (Remarks to the Author):

Dear editor and authors,

I have read the new version of the manuscript. The text is now more clear and better
focuses on the coalescing event and its modelisation. Overall, the paper now is in
good shape for publication.

I have a few minor comments below (line number corresponding to the new manuscript
without tracked changes)

**Comment 4:** line 150-152: I am still not completely convinced that there are “two
distinct peaks in tremor counts , with an additional one”... maybe you can annotate the
figure to show these 3 peaks in tremor counts...

**Response:** Following your suggestion, we added three purple triangles at the relevant
positions, as shown in Figure R1, along with the corresponding figure captions.

*Fig. R1 The moment rate and statistics of ETS in 2021 and the evolution of SSE2. a,*
 *Moment rate history (blue curve), and daily near-field (within 50 km) and total number*
 *of tremors (red and yellow histograms, respectively) during the whole ETS. The vertical*

*dash lines indicate the coalescence time of SSEs. The gray shaded area is used to*
*distinguish the two SSEs. The two black triangles at the top of the x-axis indicate the*
*two peak values of the moment for the SSE2. The three purple triangles at the down of*
*the x-axis indicate the three increases of the tremors for the SSE2. **b**, Spatio-temporal*
*evolution of SSE2. Daily slip rate contoured in color maps. The orange contour lines*
*indicate the cumulative slip area every 3.5 mm of SSE1. The black dots represent the*
*corresponding daily tremor locations from the Pacific Northwest Seismic Network*
*(PNSN) catalog (<https://pnsn.org>). The black dashed lines indicate depths at 20 km*
*interval.*

**Comment 5:** Fig 2: notation SSE 1 and SSE2 correspond to what you name before
SSE2 and SSE2b.

**Response:** We have defined it in lines 106-108:

*"Based on the spatial continuity in the evolution of the slip distribution, we segmented*
*the event into two phases: slip activity preceding April 14 (designated SSE1) and the*
*subsequent phase (designated SSE2) ."*

**Comment 6:** Fig3: (h): the time axis is not aligned to the one of figure 3(e), while it is
the case for the other models.

**Response:** Thank you very much for your thorough review. We have aligned the axes
of Figure 3(e).

**Comment 7:** Fig3: you should explain somewhere the cause of the strong
variations/oscillations)of the stress curves in fig g, h and i (because they look like noise).

**Response:** The maximum stress path propagates along the strike direction, but due to
the resolution of the patch size, the patch with the maximum stress does not move
continuously. This causes fluctuations in the curve of the maximum stress. We have
added the reason for these fluctuations in the figure caption (Lines 605-607):

*"Due to the resolution of the patches setting, the measured maximum stress exhibits*
*oscillations."*

**Comment 8:** line 260 there is an sign = that should not be there

**Response:** Done.

**Comment 9:** line 286: indicate unity for T and M0

**Response:** I guess you're referring to the units for T and M_0 . We add the corresponding
units (Lines 255-256):

*"where T is the duration (in units of seconds), M_0 is the event moment (in units of $N m$)*
*defined as the integral of slip over the rupture area multiplied by the shear modulus."*

**Comment 10:** line 290-292: "Whether there is a kink in the moment-duration scaling
law when SSEs start to migrate, the catalogue does not show it or it is not resolvable."
The grammar of the sentence sounds weird to me, but I am not native english speaker.

**Response:** Thank you for pointing out the lack of clarity in this section. Reviewer 1's
suggestion has addressed the issue we described here. We performed separate fits for
the two populations (migrating SSEs and non-migrating SSEs), and the results showed
that the c-values for the two populations are 1.5 and 3.6, respectively, which can be
explained using the bounded rupture model. For more details, please refer to **Comment**
**3.**

**Comment 11:** line 292 "energy" :you measure moment not energy, so it may be better
touse "moment"

**Response:** Done.

**Comment 12:** line 388: green function -> Green function

**Response:** Done.

**Comment 13:** fig S24b: caption and/or axis are not clear to me > what is the value
indicated as "time (s)" ?

**Response:** The x-axis label represents the duration of SSEs, so it is labeled as "time
(s)." We have provided a more detailed explanation in the figure caption to avoid any
potential misunderstanding by the readers:

*"b, Probability density distribution in the duration ($\log_{10}[\text{duration}(s)]$) and moment*
*($\log_{10}[\text{moment}(s)]$) of common SSEs between the two catalogs."*

**Response to Reviewer 3' Comments:**

Reviewer #3 (Remarks to the Author):

The authors have significantly clarified the message of the paper and the connections
between the different parts. I still have a few remarks and comments that could be
clarified before publication.

**Comment 14:**New figure S1 and discussion on coalescing events

I do not completely agree with the features and discussion presented in the new
additional figure S1. The authors claim that the coalescence of SSE in the deceleration
phase requires at least one migrating event.

It seems to me that all SSEs that reach a certain size (when they fill the width of the
SSE zone) start to migrate. This is what is shown in Fig. 4, where all large – and only
large SSEs are migrating. I thus do not fully agree with the sketch presented on Figure
S1:

The difference in size between migrating / non migrating SSE is not shown here, and the
aspect ratio between the circular growth versus along strike propagation are not clearly
represented. While it is not required for a sketch to be at scale, I think this lack of scaling
is misleading, as some of the scenarios presented might not be detectable from a geodetic
point of view: I'm not sure that the case "a" where two SSEs are merging during the
acceleration phase without migration would be detectable from a geodetic point of view:
for two non-migrating SSEs to coalesce, this would require their initiation to be very
close (less than the width of the SSE zone, assuming that the SSE initially grow as a
circle. Do you think geodetic inversion which includes some spatial smoothing, would
be able to robustly separate initiation points that are very close? It is misleading to
represent the same distance between initiation points in case "a" and "c" in Fig. S1, as
the distance would need to be much closer in case "a" than in case "c" for the events to
coalesce.

Do you have examples of non-migrating coalescing SSEs? Unless I am mistaken, in
all examples discussed in this paper, including the 2013 from Bletery and Noquet, SSEs
are always migrating before the coalescence. Does case "a" represent a realistic
scenario?

**Response:** We appreciate the reviewer's careful reading and the constructive points
regarding Figure S1. We believe part of the disagreement may stem from our definition
of "migration." As clarified in the lower-left corner of Fig. S1, we specifically define
migration as the movement of the rupture centroid of SSEs along strike. In contrast,
SSEs that expand until they reach the slip zone boundaries and then continue rupturing
primarily along strike are not considered migrating in our definition, because their
rupture centroid remains essentially stationary. We provide a clearer definition in lines
248-250:

*"We categorize SSEs in the catalog into migrating SSEs and non-migrating SSEs by
examining whether centroid of SSE ruptures and tremors migrate along the strike."*

From this perspective, we categorize the 2013 SSEs discussed by Bletery and Noquet
as non-migrating coalescing SSEs. Although those events did reach the slip zone

boundaries and subsequently ruptured along strike until coalescence, their rupture
centroid did not move significantly from nucleation to termination. This is why we
place them in the non-migrating category in Fig. S1.

We agree that aspect ratio can be misleading if interpreted as a scaled representation.
However, our intent in Fig. S1 is not to convey geometrical scaling but rather to
emphasize the distinction between coalescence scenarios with and without rupture-
center migration. We therefore think the key message of Fig. S1 lies in this definition,
rather than in exact geometric proportions.

**Comment 15:**Regarding the numerical models presented:

It is stated in the text (l.235-236) that models without frictional heterogeneities can also
generate migrating SSEs. It would be interesting to show the result of such a model. If
frictional heterogeneities are not required to generate coalescing SSEs, the authors need
to better explain why they choose to add additional complexity to generate those events:
if they are indeed more frequent with the addition of frictional heterogeneities, this
could be discussed (with some statistics on the number of coalescing events).

Indeed, the author show one example of coalescing SSE for each model (I, II or III). It
could be interesting to discuss how representative these models are, and what are the
probability of observing migrating SSEs in each model: are some frictional
configurations more likely to generate coalescing SSEs ? Are they frequent or rare ?
Could you quantify the probability of observing coalescing SSEs in each scenario ?
How does it compare with natural observations with natural observations ?

**Response:** First, we would like to clarify the necessity of the heterogeneous model for
the following reasons:

- 1. Geodetic observation data (tremors and SSEs) indicate that the fault properties in
this region are not homogeneous. A homogeneous numerical simulation model
cannot accurately represent the fault characteristics of this area.
- 2. While a homogeneous model setup can produce coalescence events, we did not
observe a decelerating coalescence of SSEs. Additionally, the coalescence
frequency in the homogeneous model is higher than what we observed (Fig. S18).

Second, regarding the impact of heterogeneity on the coalescence frequency, we briefly
discussed this in lines 214-215 of the manuscript: *Nevertheless, as ΔD_c and $\Delta \sigma_n$*
*increase, they might act as an actual barrier, thus preventing or reducing the*

*occurrence of SSE coalescence (Fig. S21). Fig. S21 provides the statistical frequency*
*of SSE coalescence as ΔDc and $\Delta\sigma_n$ increase.*

Minor comments:

**Comment 16:-** 1.76-79: Do you really want to refer to the acceleration / deceleration of
the rupture front ? I don't think this can be easily resolved, and it is not discussed in the
paper of Bletery&Nocquet. I think you should rather discuss acceleration / deceleration
in slip rate and/or moment rate.

**Response:** The description in lines 76-79 is based on the experiment by Latour, S. et
al.¹ . The change in the rupture front is related to the rupture area, which affects the
change in moment. We think that describing the variation of the rupture front may be
more helpful for readers to intuitively understand.

**Reference:**

(1) Latour, S., Passelègue, F., Paglialunga, F., Noël, C. & Ampuero, J.p. What happens
when two ruptures collide? Geophysical Research Letters 51, e2024GL110835
(2024).

**Comment 17:-**l. 240: “and event’s duration” => could be clarified by “and increasing
the event duration”

**Response:** Done.

Dear editor and authors,

I have read the new version of the manuscript. The text is now more clear and better focuses on the coalescing event and its modelisation. Overall, the paper now is in good shape for publication.

I have a few minor comments below (line number corresponding to the new manuscript without tracked changes)

- line 150-152: I am still not completely convinced that there are “two distinct peaks in tremor counts , with an additional one”... maybe you can annotate the figure to show these 3 peaks in tremor counts...
- Fig 2: notation SSE 1 and SSE2 correspond to what you name before SSE2a and SSE2b...
- Fig3: (h): the time axis is not aligned to the one of figure 3(e), while it is the case for the other models.
- Fig3: you should explain somewhere the cause of the strong variations/oscillations) of the stress curves in fig g, h and i (because they look like noise).
- line 260 there is an sign = that should not be there
- line 286: indicate unity for T and M_0
- line 290-292:”Whether there is a kink in the moment-duration scaling law when SSEs start to migrate, the catalogue does not show it or it is not resolvable.” The grammar of the sentence sounds weird to me, but I am not native english speaker.
- line 292 “energy” :you measure moment not energy, so it may be better to use “moment”
- line 388: green function -> Green function
- fig S24b: caption and/or axis are not clear to me > what is the value indicated as “time (s)’ ?